# Single cell multi-omics reveal intra-cell-line heterogeneity across human cancer cell lines

Qionghua Zhu [1,2,14] ✉, Xin Zhao[3,4,14], Yuanhang Zhang[3,4,14], Yanping Li[2,14], Shang Liu[3,14], Jingxuan Han[2], Zhiyuan Sun[2], Chunqing Wang[3,4], Daqi Deng[2], Shanshan Wang[3], Yisen Tang[2], Yaling Huang[3], Siyuan Jiang[3,4], Chi Tian [2], Xi Chen[3], Yue Yuan[3], Zeyu Li [3,4], Tao Yang [5], Tingting Lai[5], Yiqun Liu[5], Wenzhen Yang[5], Xuanxuan Zou[3,4], Mingyuan Zhang[3], Huanhuan Cui [1,2,6], Chuanyu Liu [3], Xin Jin [3], Yuhui Hu [1,2,7], Ao Chen [3,8,9], Xun Xu [3], Guipeng Li [1,2,6], Yong Hou[3,10], Longqi Liu [3,11,12] ✉, Shiping Liu [3,9,10,11,12] ✉, Liang Fang [1,2,6] ✉, Wei Chen [1,2] ✉ & Liang Wu [3,8,13] ✉

Human cancer cell lines have long served as tools for cancer research and drug discovery, but the presence and the source of intra-cell-line heterogeneity remain elusive. Here, we perform single-cell RNA-sequencing and ATAC-sequencing on 42 and 39 human cell lines, respectively, to illustrate both transcriptomic and epigenetic heterogeneity within individual cell lines. Our data reveal that transcriptomic heterogeneity is frequently observed in cancer cell lines of different tissue origins, often driven by multiple common transcriptional programs. Copy number variation, as well as epigenetic variation and extrachromosomal DNA distribution all contribute to the detected intra-cell-line heterogeneity. Using hypoxia treatment as an example, we demonstrate that transcriptomic heterogeneity could be reshaped by environmental stress. Overall, our study performs single-cell multi-omics of commonly used human cancer cell lines and offers mechanistic insights into the intra-cell-line heterogeneity and its dynamics, which would serve as an important resource for future cancer cell line-based studies.

The evolution of cancer starts with malignant transformation, followed by progression to more aggressive and resistant forms, toward poor clinical outcomes. The intratumoral heterogeneity in human tumors plays a critical role in carcinogenesis and tumor evolution[1]. In recent years, single-cell genomics has made incredible progress toward characterizing specific tumor subtypes and disentangling the cellular complexity of a given tumor[2,3]. However, most of the effort has been so far focused on profiling the clinical samples.

[1]Shenzhen Key Laboratory of Gene Regulation and Systems Biology, School of Life Sciences, Southern University of Science and Technology, 518055 Shenzhen, China. [2]Department of Systems Biology, School of Life Sciences, Southern University of Science and Technology, 518055 Shenzhen, China. [3]BGI Research, 518083 Shenzhen, China. [4]College of Life Sciences, University of Chinese Academy of Sciences, 100049 Beijing, China. [5]China National GeneBank, 518120 Shenzhen, China. [6]Academy for Advanced Interdisciplinary Studies, Southern University of Science and Technology, 518055 Shenzhen, China. [7]Department of Pharmacology, School of Medicine, Southern University of Science and Technology, 518055 Shenzhen, China. [8]JFL-BGI STOmics Center, Jinfeng Laboratory, 401329 Chongqing, China. [9]The Guangdong-Hong Kong Joint Laboratory on Immunological and Genetic Kidney Diseases, Guangdong, China. [10]Shenzhen Key Laboratory of Single-Cell Omics, BGI-Shenzhen, 518100 Shenzhen, China. [11]BGI Research, 310012 Hangzhou, China. [12]Shenzhen Bay Laboratory, 518000 Shenzhen, China. [13]BGI Research, 401329 Chongqing, China. [14]These authors contributed equally: Qionghua Zhu, Xin Zhao, Yuanhang Zhang, Yanping Li, Shang Liu. ✉e-mail: zhuqh@mail.sustech.edu.cn; liulongqi@genomics.cn; liushiping@genomics.cn; fangl@sustech.edu.cn; chenw@sustech.edu.cn; wuliang@genomics.cn

Yet, the molecular understanding of what drives such heterogeneity remains elusive, which necessitates extensive follow-up studies in different model systems. Human cancer cell lines have long served as tools for cancer research and drug discovery, as they represent key properties of their original tumors and provide context for studying cellular or molecular mechanisms and testing therapy responses. Given the establishment of cancer cell lines involves the selection of tumor cells that adapt to in vitro culture conditions, cancer cell lines were often thought to be homogenous and unable to maintain the heterogeneity of the original tumor. However, it has been shown that cell lines could evolve and develop into distinct cell line strains[4]. Many recent studies have demonstrated that cellular diversity could be detected in established cell lines by using techniques including single-cell RNA-sequencing (scRNA-seq) and fluorescence-activated cell sorting (FACS)[5-7]. Transcriptome heterogeneity within cell lines has facilitated the discovery of critical regulators that drive the epithelial-mesenchymal transition (EMT) program and drug resistance in subclones[5,8]. If certain intra-cell-line heterogeneity is associated with therapeutic response, these cell lines could potentially serve as suitable models to study the molecular mechanisms underlying drug sensitivity and to test different treatment schemes. Therefore, understanding the heterogeneity of commonly used human cancer cell lines will not only provide important information for cell line-based biomedical research but also may help to identify suitable models for investigating novel mechanisms underlying specific cancer phenotypes. More recently, Kinker et al. systematically profiled 198 cancer cell lines by scRNA-seq, which mainly consisted of those derived from lung cancer, and revealed the landscape and recurring patterns of intra-cell-line heterogeneity. However, the molecular mechanisms underlying the heterogeneity still await to be uncovered.

The origin of intratumoral heterogeneity was traditionally considered in genetic terms but has been extended into more facets, including epigenetics and dramatic changes in the microenvironment[9-11]. Although effective at resolving the heterogeneity in transcriptional programs, scRNA-seq is unable to reveal the underlying driving force, such as epigenetic factors. For this purpose, it requires techniques that capture cell-to-cell variations in their epigenetic landscape. Since chromatin accessibility to a large extent could reflect transcription factor binding, histone modifications and DNA methylation, and offers greater insights into the gene regulatory mechanisms, single-cell sequencing assay for transposase-accessible chromatin (scATAC-seq) has recently become the most widely used assay for epigenomic profiling at single-cell resolution. A major advantage of scATAC-seq, compared to scRNA-seq, is that it provides mechanistic insights into gene regulation modulated by transcription factors[12]. The joined application of scRNA-seq and scATAC-seq to cancer specimens helps to discern precise cis-regulatory elements and target genes as well as identify the key regulatory networks that govern tumor development.

To characterize both transcriptomic and epigenetic heterogeneity within different cell lines, we perform scRNA-seq and scATAC-seq on dozens of human cell lines, mainly consisting of breast and colorectal origins. By integrating the scRNA-seq and scATAC-seq data, we investigate the molecular mechanisms that drive heterogeneity and find that copy number variation (CNV) only contributes partially to the observed transcriptomic heterogeneity. Epigenetic diversity and extrachromosomal circular DNA (ecDNA) distribution contribute significantly to the intra-cell-line heterogeneity. Moreover, through lineage tracing and hypoxia treatment, we demonstrate that the transcriptomic heterogeneity is plastic and could be reshaped under environmental stress. Taken together, our study performs single-cell multi-omics of commonly used human cancer cell lines and offers mechanistic insights into the intra-cell line heterogeneity.

## Results

### Pan-cancer scRNA-seq of human cell lines

In this study, 40 human cancer cell lines, which were distributed among 9 lineages and dominated by solid tumors, and 2 human normal cell lines were selected for scRNA-seq profiling (Fig. 1a, Supplementary Table 1). For mammary and colorectal cancer, 23 cell lines of different molecular subtypes were selected (Supplementary Table 2). To increase the throughput and reduce the cost of scRNA-seq experiments, three cell lines from different lineages were pooled for each scRNA-seq run and then computationally assigned to the corresponding cell line according to their expression features (Fig. 1b and Supplementary Fig. 1a, b; see "Methods"). The effectiveness of our assignment analysis was then further validated (Supplementary Fig. 1c–e, see "Methods"). We obtained a total of 23,089 cells with an average of 513 cells per cell line, 34,641 transcripts (represented by unique molecular identifier, UMI), and 5859 genes captured per cell, underscoring the high quality of our dataset (Supplementary Fig. 1f–h). The correct assignment of individual cell lines was again confirmed by matching the scRNA-seq profile with the bulk RNA-seq profile generated by the Cancer Cell Line Encyclopedia (CCLE) or GEO[13-15] (Supplementary Fig. 1i). To combine each batch of scRNA-seq data and generate a single-cell transcriptomic encompassing all 42 cell lines, Seurat[16] was used and each of 42 cell lines formed a distinct cluster in the Uniform Manifold Approximation and Projection (UMAP) (Fig. 1c and Supplementary Fig. 1j; available also online: https://db.cngb.org/cdcp/scatlashcl/). The cluster distribution was not associated with read counts or sequencing batches (Supplementary Fig. 1k, l). To further examine the reproducibility of our scRNA-seq experiment, we analyzed three cell lines in two independent experiments, including Caco-2, SCC-4, and MDA-MB-231. The results showed that cells of the same cell line, but measured in two independent experimental runs, were intermingled (Fig. 1c and Supplementary Fig. 1m).

To gain further insights into specific cancer lineages, we plotted breast and colorectal cancer cell lines in UMAP, separately. For breast cancer cell lines, cells of the same molecular subtypes, including luminal A (LA), luminal B (LB), Her2+ (H), triple-negative A (TNA), and triple-negative B (TNB), were adjacent to each other (Fig. 1d). The expression of clinically relevant biomarkers, including *ESR1* and *ERBB2* (*HER2*), were in consistence with their reported status[17]. For instance, *ESR1* was mainly expressed in LA and LB subtypes, while *ERBB2* was upregulated in LB and H subtypes (Fig. 1e). Moreover, we analyzed the expression of additional well-known stromal and epithelial markers of clinical relevance (Fig. 1f)[18,19]. Luminal and Her2+ cell lines highly expressed epithelium genes (*EPCAM*, *CDH1*, etc.), but neither basal epithelial (*KRT14*, etc.) nor stromal markers (*VIM*, etc.), whereas triple-negative cell lines expressed high levels of basal cell marker genes, and TNB cell lines further showed high expression levels of stromal and epithelial-mesenchymal transition (EMT)-related genes (*FN1*, etc.). In colorectal cancer, however, cells of the same subtype were not completely clustered together, in particular for those of CMS1 type (Supplementary Fig. 1n). For instance, in UMAP projection HCT-15 and DLD-1 cells, which were previously defined as CMS1 subtype, were separated from another two CMS1 subtype cell lines[20]. Considering these cell line subtypes were previously defined based on microarray data[20], we employed the same R package to redefine the cell line subtypes based on RNA-seq data in the CCLE or GEO database. As a result, most of the cell lines remained in their original subtype, but DLD-1 was clustered as CMS3 subtype (Supplementary Table 2). Notably, Sveen et al. have also shown that HCT-15 and DLD-1 could not be confidently assigned to any subtype[21], suggesting that these cell lines might not be suitable for research strictly requiring CMS1 or CMS3 subtype.

### Transcriptomic heterogeneity within individual cell lines

We then zoomed into the single-cell transcriptome profile of individual cell lines. According to the pattern of transcriptome heterogeneity

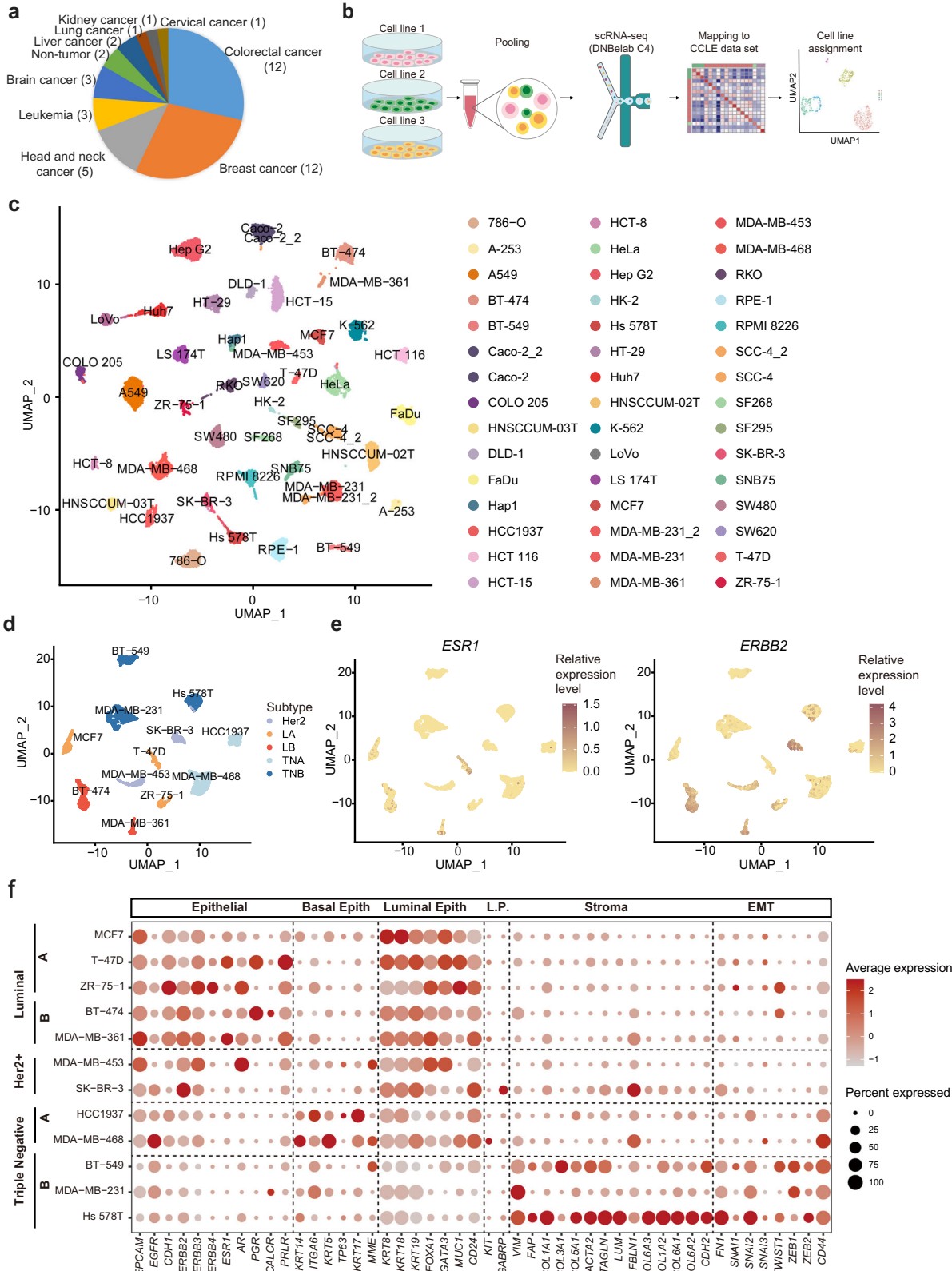

**Fig. 1 | Characterizing cellular heterogeneity within cell lines by scRNA-seq.**
**a** Quantitative scRNA-seq analysis of 42 diverse cell lines from 9 lineages. The cell line number of each lineage is indicated. **b** Schematic of the experimental workflow of scRNA-seq analysis. Three cell lines were pooled for scRNA-seq and then data from Harmonizome was utilized to assign cells to the most similar one based on their gene expression profile. **c** UMAP plot of all cell lines demonstrating the robustness of cell line assignment. Different cell lines were labeled in different colors. **d** Graphical representation of single-cell transcriptomics of breast cancer cell lines according to cancer subtype labeled in different colors. **e** Normalized expression levels of indicated biomarker genes (*ESR1* and *ERBB2*) in individual cells, with redness indicating expression level. **f** Bubble plot represents the average expression levels of marker genes and fractions of expressed cells in breast cancer cell lines. Basal Epith = Basal Epithelial, Luminal Epith = Luminal Epithelial, L.P. = Luminal Progenitor, EMT = Epithelial to Mesenchymal Transformation. Source data are provided in the Source Data file.

reflected in UMAP, 42 cell lines could be roughly divided into two types: discrete and continuous (Fig. 2a and Supplementary Fig. 2a). Whereas distinct subclusters were observed as the spatially discrete ones (e.g., Hs 578T and SNB75) likely due to the presence of subclones in these cell lines, the continuous ones showed a hairball pattern without a clear border between subclusters (e.g., A549) (Fig. 2a and Supplementary Fig. 2a). In total, 25 and 17 cell lines (57% and 43%) belong to the discrete and continuous group, respectively

(Supplementary Fig. 2b and Supplementary Table 1). Although the distinction between discrete and continuous might suggest different mechanisms underlying the intra-cell-line heterogeneity, such classification could not quantitatively reflect the level of heterogeneity within individual cell lines, which could be better represented as the spread of cells in their respective UMAP. Therefore, we used another metric, i.e., 'diversity score', to systematically quantify the intra-cell-line heterogeneity of each cell line based on scRNA-seq data[22]. Briefly,

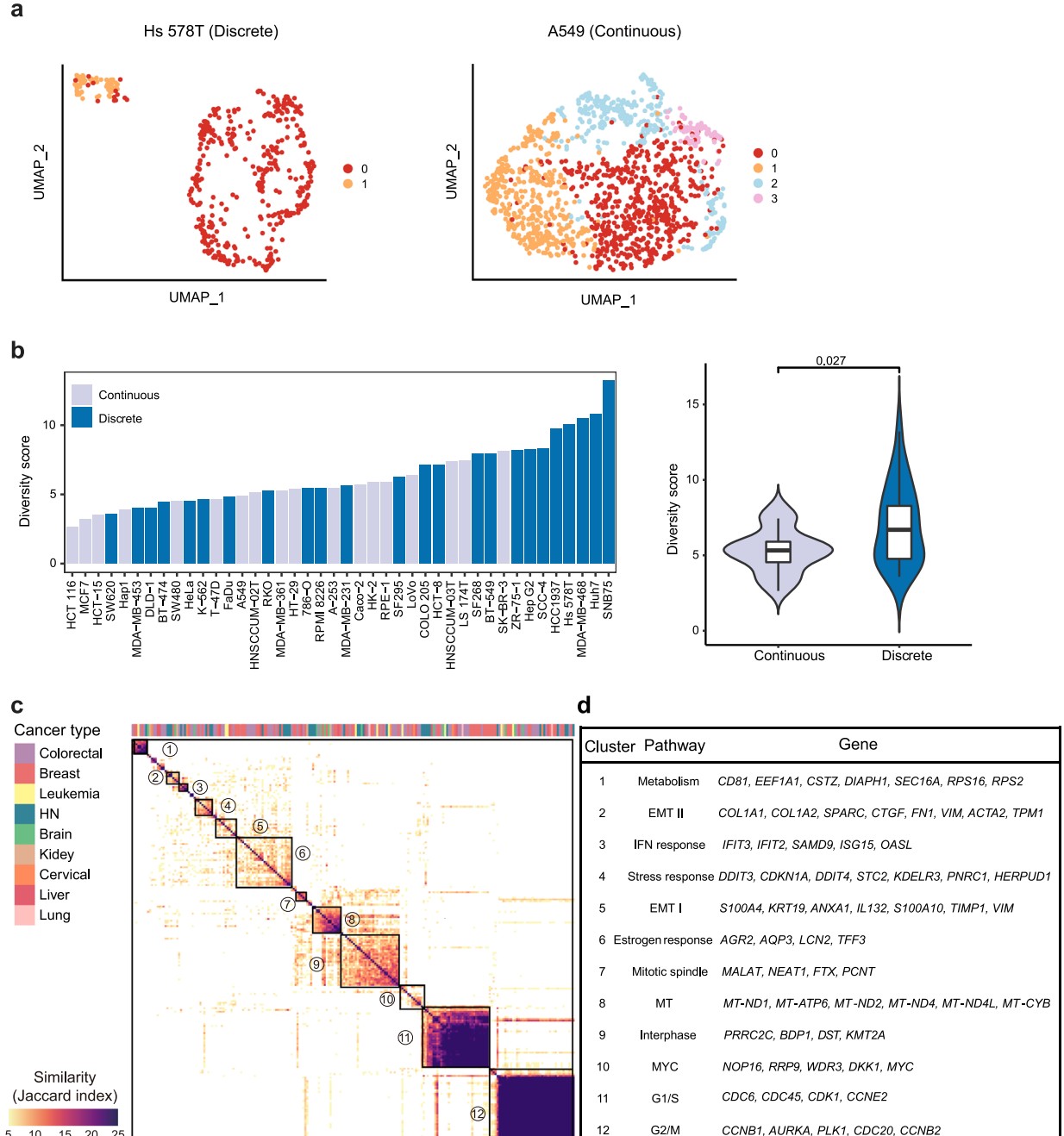

**Fig. 2 | Different patterns of transcriptome heterogeneity within cell lines.**
**a** Illustration of two types of transcriptional heterogeneity: continuous vs discrete; UMAP plot showing exemplary cell lines for different patterns: Hs 578T for discrete and A549 for continuous. **b** Diversity scores for different cell lines, cell lines with light blue indicating continuous pattern and cell lines with dark blue indicating discrete pattern. A violin plot depicted the diversity score of the cell lines with the two patterns, including continuous ($n = 18$ cell lines) and discrete ($n = 24$ cell lines). For each boxplot, the center line represents the median, the box indicates the

upper and lower quartiles and the whisker represents 1.5-fold of the interquartile range. A one-sided Wilcoxon test was used to test the statistical significance ($p$-value = 0.021). **c** The main heatmap depicts pairwise similarities between all NMF programs, quantified by Jaccard Index over the programs' genes, ordered by hierarchical clustering. Twelve clusters are indicated by squares and numbers. **d** Annotation and selected top marker genes were shown for each cluster. Source data are provided in the Source Data file.

we employed PCA to project all cells captured in scRNA-seq to an eigenvector space and defined the centroids of the individual cell line, and then calculated the diversity score as the average distance within individual cell lines to their specific centroids ("Methods"). In general, the discrete/continuous classification correlated well with the diversity score, where the majority of the cell lines with the top 25% highest diversity score were those showing discrete pattern (9/10; Fig. 2b), and the diversity score in the discrete group is significantly higher than in continuous one (Fig. 2b right panel), suggesting cell lines with a high degree of heterogeneity often derives distinct subclasses. However, there were exceptions: some discrete cell lines had low diversity scores, such as SW620, whereas continuous pattern cell lines derived high diversity scores, such as SK-BR-3(Fig. 2b). For the former, the within-cluster distances are significantly smaller than the between-cluster ones. As a result, the cell line showed a discrete pattern according to the classification, even though its diversity score is low given that the collective distance is not that large. For the latter, the collective distance can be large whereas the within-cluster distance is not significantly different from the between-cluster ones. Taken together, the intra-cell-line heterogeneity demonstrated by two metrics indicated that transcriptome diversity could be affected by the presence of distinct subclones, as well as the intrinsic plasticity of the cell line, and the contribution of these two factors to the intra-cell-line heterogeneity varies among individual cell lines.

## Molecular features shaped by transcriptomic heterogeneity

To functionally characterize the extensive variability in gene expression, we employed a computational framework, which has previously been used to determine the heterogeneous expression programs[8]. In essence, non-negative matrix factorization (NMF) was used to search heterogeneously activated expression programs (termed NMF programs) within each cell line ("Methods") (Supplementary Fig. 2c). In total, we determined 228 programs, with 2-8 programs for each cell line (Supplementary Data 1). Next, we compared the NMF programs based on their shared genes and merged similar NMFs into NMF clusters to determine recurrent NMF programs among 40 cancer cell lines. As a result, 12 clusters that were heterogeneously activated in more than 3 cell lines were identified (Fig. 2c and Supplementary Data 2). Subsequently, based on shared genes among the individual programs within the same cluster, where the genes appearing in more than 25% of programs were used as signature genes, gene set enrichment analysis (GSEA) was employed to annotate each cluster. The two most prominent program clusters, shared by the majority of cell lines (39/40) are related to the cell cycle, including the G1/S and G2/M programs (Fig. 2c, d and Supplementary Fig. 2d), suggesting that the cell cycle variation is the primary cause of the observed transcriptomic heterogeneity in most cell lines. Besides cell cycle-related programs, another 10 program clusters represent various critical biological processes (Fig. 2d), largely independent of cell cycle status (Supplementary Fig. 2e). For instance, program cluster 1 consists of genes related to protein metabolisms, such as *CD81*, *EEF1A1*, *CSTZ*, *UPF1*, *RPS16*, and *PRS2*. Program clusters 2 and 5 are both associated with the EMT, but with different marker genes and reflect EMT-like processes in distinct cell lines. Program cluster 2 (EMT II) includes *SPARC*, *CTGF*, *FN1*, *VIM* and other genes, which resemble the partial EMT and are related to cancer metastasis[6], while Program cluster 5 (EMT I) includes *VIM*, *S100A4*, *TIMP1*, *KRT19*, etc., which prompt the progression of different cancer via EMT[23–25]. The identification of different EMT-associated program clusters suggested EMT as a common but context-specific driver of intra-cell-line heterogeneity. Program cluster 3 contains interferon response genes, such as *IFIT1*, *IFIT2*, and *IFIT3*, whose heterogeneity has previously been observed in ovarian cancer[26]. Program cluster 4 harbors stress response genes, including DNA damage-induced genes *DDIT3* and *DDIT4*, which are reported heterogeneously activated in HNSCC[6]. Program cluster 6, which is termed as estrogen

response, was significantly enriched in breast cancer cell lines (11/12) (Supplementary Fig. 2d). Program clusters 7, 8, 9, and 10 are related to the mitotic spindle, mitochondria (MT), interphase, and MYC pathway respectively. Moreover, we further examined the co-occurrence of these programs within cell lines. As expected, two EMT program clusters (program 2 and 5) are significant co-occurrences in cell lines and EMT II also coexists with the IFN response program (Supplementary Fig. 2f). Besides, the 'Interphase' tends to coincide with 'Mitotic spindle' and 'MT'. These results indicated that some heterogeneously activated program clusters may be entangled with each other or result from common upstream factors.

Additionally, we compared program clusters identified in our study with a recent study, where 198 cancer cell lines were analyzed by scRNA-seq, and 12 programs were determined[8]. According to the similarity of signature genes, six out of twelve program clusters were identified in our study, including IFN response, stress response, EMT I, EMT II, G1/S and G2/M (Supplementary Table 3). This suggests that even with largely diverged transcriptome features, the heterogeneous transcriptional programs were shared among different cancer cell lines of different tissue origins. Moreover, due to the different spectrum of cancer cell lines that were profiled in two studies, some recurrent expression program clusters were only observed in our analysis, such as estrogen response (Program 6), which was enriched in breast cancer cell lines (Supplementary Fig. 2d).

## Pan-cancer scATAC-seq of human cell lines

Although single-cell transcriptomics allows the detection of cellular diversity, it remains a challenge to interpret the epigenomic regulatory code based solely on the transcriptome data. By comparison, scATAC-seq provides a more direct characterization of the genome-wide activity of enhancers and promoters in heterogeneous cell populations[27]. Therefore, we further used scATAC-seq to investigate the epigenomic heterogeneity within individual cell lines at single-cell resolution. To increase the throughput and reduce the cost of scATAC-seq experiments, at most six cell lines with distinct expression profiles were pooled for each scATAC-seq run and then computationally assigned to the corresponding cell line according to the accessibility features of the differentially expressed genes (Fig. 3a and Supplementary Fig. 3a).

We utilized a framework that has been used to identify cell clusters in different types of samples to analyze scATAC-seq data[28]. First, after quality filtering, 54,597 high-quality single nuclei were obtained, with a median of 1170 nuclei (ranging from 203 to 4391 nuclei) per cell line and median fragments of 11,702 (ranging from 1000 to 99,444) per nucleus (Supplementary Fig. 3b, c). Then, we aggregated the scATAC-seq profile to calculate the genomic distribution of open chromatin regions, which revealed that 35.5% of ATAC peaks were localized in promoter-proximal regions (<±1.5 kb TSS). We then used UMAP to plot 39 cell lines together to present the inter-cell-line distance at the chromatin level and each cell line formed a distinct cluster (Fig. 3b).

To assess the intra-cell-line heterogeneity at the epigenomic level, we partitioned cells into preliminary clusters within cell lines. In contrast to scRNA-seq data, the scATAC-seq data was highly sparse and therefore it was challenging to define distinct cell clusters based on scATAC-seq data using UMAP, even for cell lines harboring clusters based on scRNA-seq data. Therefore, according to the shape of the cell population projected in UMAP, cell lines were roughly classified into two categories: indiscriminate and differential ones. Cells in the indiscriminate ones showed no obvious clustering pattern (e.g., COLO 205), while distinct clusters were observed in the differential ones (e.g., MDA-MB-231 and SF268). As a result, 62% of cell lines showed an indiscriminate pattern, while 38% showed a differential pattern (Fig. 3c, Supplementary Fig. 3d and Supplementary Table 4). Interestingly, among cell lines with the differential pattern, only half of them (7/15)

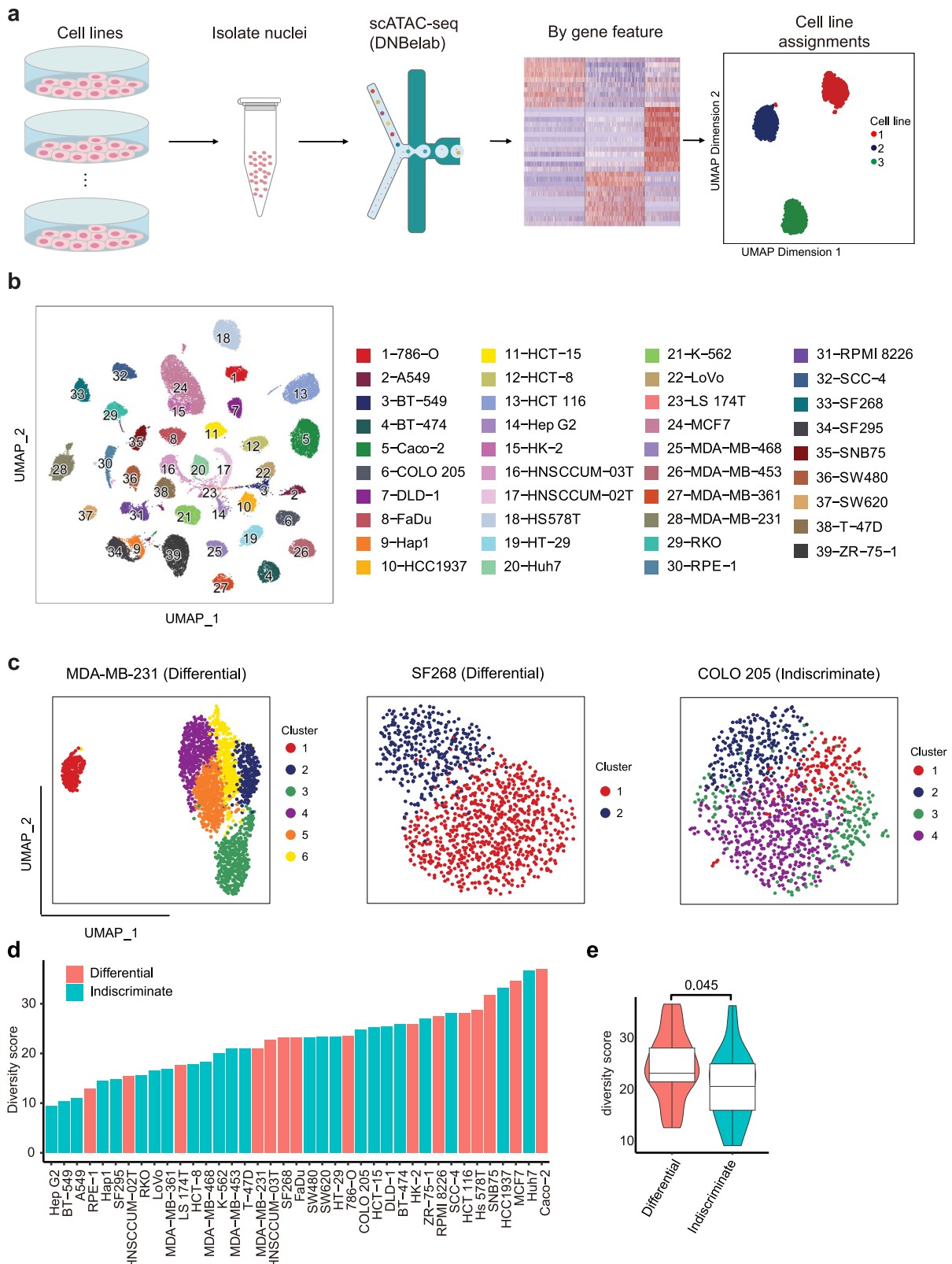

exhibited discrete pattern determined by scRNA-seq data. This result suggested that scRNA-seq and scATAC-seq could reveal cellular heterogeneity at different regulatory levels. Meanwhile, we also applied a diversity score to quantify the intra-cell-line heterogeneity of each cell line based on scATAC-seq data. Similar as done for scRNA-seq data, we employed PCA from ArchR analysis result to project all cells captured in scATAC-seq to an eigenvector space and defined the centroids of the

individual cell line, and then calculated the diversity score as the average distance within individual cell lines to their specific centroids. In general, the indiscriminate/differential classification correlated well with the diversity score, and the diversity score of cell lines belonging to the differential group was significantly higher than those from indiscriminate groups (Fig. 3d, e), suggesting cell lines with a high degree of heterogeneity often derives distinct subclusters.

**Fig. 3 | Characterizing the heterogeneity of chromatin accessibility within cell lines by scATAC-seq. a** Schema of the experimental workflow of scATAC-seq analysis. Cell lines were pooled for scATAC-seq and then differential genes were utilized to assign cells to the most similar cell line based on the gene score calculated according to their chromatin accessibility. **b** UMAP representation of all cell lines' scATAC profile. Different cell lines were labeled in different colors. **c** UMAP plot illustration of two types of expression variability: with (differential, MDA-MB-231 and SF268) or without (indiscriminate, COLO 205) distinct chromatin

accessibility. **d** Diversity scores for different cell lines. Cell lines with light blue indicate an indiscriminate pattern and cell lines with red indicate a differential pattern. **e** A violin plot depicted the diversity score of the cell lines with the two patterns, including differential (n = 15 cell lines) and indiscriminate (n = 25 cell lines). For each boxplot, the center line represents the median, the box indicates the upper and lower quartiles and the whisker represents 1.5-fold of the inter-quartile range. A one-sided Wilcoxon test was used to test the statistical significance (p-value = 0.045). Source data are provided in the Source Data file.

## Transcriptomic heterogeneity associated with CNVs

To identify the origin of observed transcriptional heterogeneity, we first turned to genetic variation, which is commonly observed in malignant tumors. For this purpose, the large-scale copy number variations (CNVs) were inferred based on scRNA-seq data using inferCNV[29], where we calculated the average expression levels in windows of 400 genes around each locus in comparison to a set of reference normal cells (RPE-1 and HK2) ("Methods"). The analysis identified CNV subclones within 25 of 40 (62.5%) cancer cell lines (Fig. 4a, b and Supplementary Fig. 4): 8 (20%) cell lines showed the match between transcriptional subclusters and CNV-based subclones (Fig. 4b upper and Supplementary Fig. 4a), which suggested that CNV induced copy number variation considerably contributes to the transcriptional variation; 17 (42.5%) cell lines showed no association (Fig. 4b middle and Supplementary Fig. 4b). Among these 25 cell lines, 18 and 7 cell lines showed discrete and continuous pattern, respectively, while, for the rest 15 cell lines, 6 and 9 cell lines showed discrete and continuous pattern, respectively (Supplementary Table 1). It suggests that cell lines tend to present a heterogeneous transcriptomic activity in a more discrete manner when genetic variation contributes to intra-cell-line heterogeneity. Indeed, among cell lines with a discrete pattern, 78.2% of cell lines had CNV subclones, and, in 30.4% of these cell lines, CNV subclones were matched to transcriptional subclusters (Fig. 4a middle). We then investigated whether the genes located in CNV regions contribute to the transcriptomic variability in these matched cell lines. The results showed that the differentially expressed genes are indeed significantly enriched in CNV regions (Fig. 4c), suggesting that genetic variation is an important factor determining the intra-cell-line heterogeneity for discrete pattern cell lines. However, the fact that not all discrete pattern cell lines have CNV subclones and CNV subclones are not always linked to transcriptional subclusters also suggests that even for those with distinct patterns there are other factors influencing the transcriptomic heterogeneity.

## Transcriptomic heterogeneity modulated by chromatin accessibility

In addition to genetic heterogeneity, accumulating data suggested that different cellular states are also encoded and propagated epigenetically. To explore the mechanisms underlying epigenomic heterogeneity, first, we applied two approaches, including ArchR[28] and ChromVAR[30], to seek to infer potential transcription factors (TFs) that drive the differential chromatin accessibility between cell subclusters and focused on common TFs that appeared in at least three cell lines (Fig. 5a and Supplementary Data 3, see "Methods"). TFs such as *IRF1*, *IRF2*, *IRF9*, and *STAT2*, which were involved in regulating the IFN response, were heterogeneously activated within LS 174T and HNSCCUM-02T cell lines. The analysis based on scRNA-seq data also showed that the IFN response transcription program heterogeneously activated in LS 174T and HNSCCUM-02T (Supplementary Data 2). Another group of common TFs including *EGR3*, *ELF1*, *KLF4*, and *TFAP2C*, which were associated with the estrogen response, were heterogeneously activated in ten cell lines. These ten cell lines, except for HT-29, K-562, and SW480, also showed heterogeneity in the Estrogen response program based on scRNA-seq data (Supplementary Data 2). These results suggested that scATAC-seq data could help to reveal the TFs responsible for the transcriptional heterogeneity.

Then, we focused on seven cell lines that exhibited discrete and differential patterns in scRNA-seq and scATAC-seq data, respectively, intending to match subclusters between the two datasets. The gene score matrix was calculated based on scATAC-seq data, which was then used to align cells from scATAC-seq to cells from scRNA-seq according to the similarity between the gene score and gene expression measured by scRNA-seq[28]. As a result, in three out of seven cell lines, including MDA-MB-231, RPMI 8226, and SNB75, cell subclusters could be confidently matched between the two datasets (Fig. 5b–d). MDA-MB-231 cells were clustered into four and three subgroups according to scATAC-seq and scRNA-seq data, respectively, where subgroup 2 of the scATAC-seq dataset was matched to a mix of subgroup 1 and 2 of the scRNA-seq dataset (Fig. 5b–d left). RPMI 8226 cells were divided into subgroup 0 and 1 based on scRNA-seq (Fig. 5c middle), while subgroup 1 was further divided into three subclusters according to scATAC-seq (Fig. 5b, d middle). For SNB75, subgroup 3 of scATAC-seq data was matched to a mix of subgroup 0 and 1 of scRNA-seq data, the remaining subgroups of scATAC-seq data showed a one-to-one correlation with scRNA-seq subgroups (Fig. 5b–d right). Detection of increased open chromatin could be attributed to either increased chromatin accessibility or increased DNA copy numbers. To exclude the influence caused by the latter, i.e., CNVs, we demonstrated that the chr15 and chr9, which showed CNV in subclusters of SNB75 and MDA-MB-231 respectively, did not show evident heterogeneity in chromatin accessibility among different subclusters (Supplementary Fig. 5a, b). This suggested that heterogeneous chromatin accessibility and CNV could independently contribute to transcriptomic heterogeneity, and for these three cell lines, chromatin accessibility contributes considerably to transcriptomic heterogeneity.

To further explore the molecular mechanisms that drive heterogeneous chromatin accessibility and RNA expression, we sought to infer potential TFs that bind to differentially accessible chromatin regions. We applied chromVAR[30], a package designed for analyzing sparse scATAC-seq data to infer TF activity by measuring the gain or loss of chromatin accessibility within sets of genomic features while controlling for technical biases, to reveal the potential TFs showing variability among subclusters (See "Methods", Fig. 5e and Supplementary Data 4). To further select confident TFs, we applied three criteria: (1) the accessibility of TF-specific binding motifs differed between subgroups; (2) the expression of TF target genes, derived from the regulons identified through the SCENIC workflow, contribute to the transcriptomic heterogeneity within the cell line; (3) the expression or chromatin accessibility of TF should be variable among the subclusters of the cell line. In MDA-MB-231, the activity of *FOXA2* was found to be specifically upregulated in subgroup 0 (Fig. 5f), and the promoter accessibility of *FOXA2*, an indicator of its transcription, was also increased in subgroup 0 (Fig. 5g). According to the transcriptomic feature, subgroup 0 had activated expression profile related to the hallmark of 'Kras_signaling_up' (Supplementary Data 1). We then analyzed the downstream targets of *FOXA2* and discovered that they were more related to genes involved in 'Kras_signaling_up' (Fig. 5h), which were especially upregulated in subgroup 0 (Fig. 5i). Recent studies have demonstrated that *FOXA2* could cooperate with KrasG12D activation during tumor development[31,32]. Our result implies that *FOXA2* may play an important role in controlling the heterogeneity

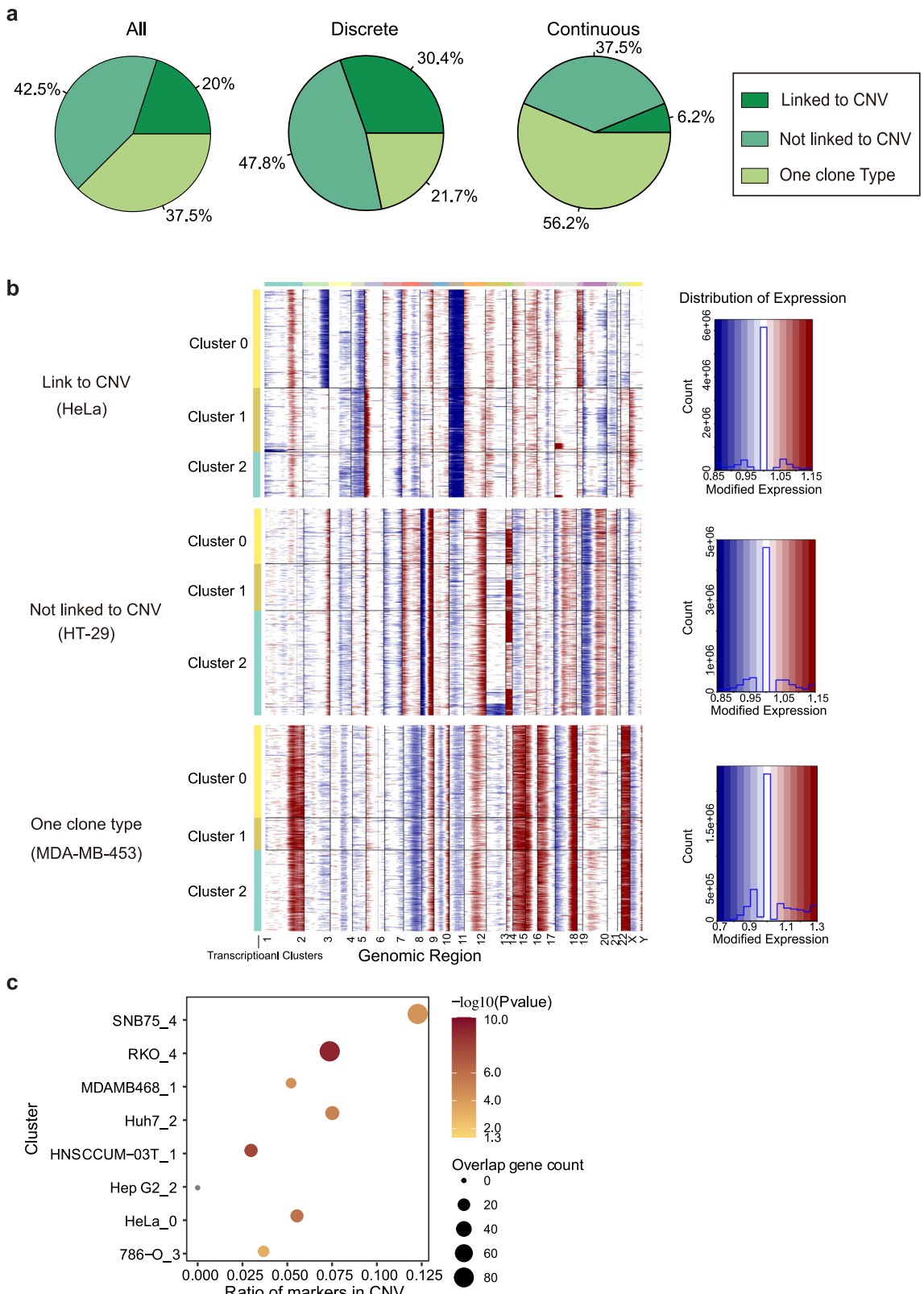

related to 'KRAS_signaling_up' within MDA-MB-231. In RPMI 8226, *NFE2L2* was upregulated and showed increased promoter accessibility in subgroup 1 (Supplementary Fig. 5c, d), which showed an activated EMT signature (Supplementary Data 1). It is in line with the fact that *NFE2L2* is a crucial regulator of tumor metastasis[33,34], and the downstream targets of *NFE2L2* were more related to genes involved in EMT regulation (Supplementary Fig. 5e, f). In SNB75, *NFYB* is a critical TF in

cluster 4 (Supplementary Fig. 5g, h). We analyzed the downstream targets of *NFYB*, which are functionally enriched for 'E2F_targets' (Supplementary Fig. 5i). These 'E2F_targets'-related genes mainly accumulated in cluster 4 (Supplementary Fig. 5j). Taken together, these results demonstrated that the combination of scRNA-seq and scATAC-seq data could facilitate the discovery of key epigenetic regulators underlying the observed cellular heterogeneity.

**Fig. 4 | Assessing the role of CNVs in cellular transcriptomic heterogeneity.**
**a** Percentage of cell lines whose transcriptomic subclusters are associated or not
associated with CNV subclones in all cell lines (left), discrete pattern cell lines
(middle), and continuous pattern cell lines (right), respectively. **b** Representative
cell lines with or without the association between transcriptomic subclusters and
CNV-based subclones: CNV subclone, identified by chromosomes 2, 5 and 18, is
linked to transcriptomic subcluster (upper, HeLa,); CNV subclones, identified by
chromosome 13, are not linked to transcriptional subclusters (middle, HT-29); there

is only one CNV clone type existing within the cell line (bottom, MDA-MB-453).
**c** The differentially expressed genes between subclusters are enriched in CNV
regions for listed cell lines (DEG number: $n = 284$ in cluster 3 of 786-O, $n = 284$ in
cluster 0 of HeLa, $n = 289$ in cluster 2 of Hep G2, $n = 159$ in cluster 1 of HNSCCUM-
03T, $n = 643$ in cluster 2 of Huh7, $n = 256$ in cluster 1 of MDA-MB-468, $n = 638$ in
cluster 4 of RKO, $n = 1351$ in cluster 4 of SNB75). A one-tailed hypergeometric test
was used to test the statistical significance. Source data are provided in the Source
Data file.

## Transcriptomic heterogeneity induced by ecDNA distribution

Recent studies showed that oncogene amplification could be found
both on chromosomes as well as extrachromosomal circular DNAs
(ecDNAs)[35,36]. Since a centromere is absent in the ecDNA, compared
with chromosomal amplicons, they are less stable and segregate sto-
chastically to daughter cells, which could potentially serve as a driving
force for cellular heterogeneity. To investigate the contribution of
ecDNA to the transcriptomic heterogeneity, we first established an
analytic method, which is able to detect highly potential ecDNAs
regions using scATAC-seq data, based on the assumptions: (1) ecDNAs
are highly amplified; (2) ecDNAs have much higher accessibility[37].
Based on the scATAC-seq data, the read coverage showed multiple
peaks in every single cell, of which the first peak was considered as the
read coverage for the genome regions of two copies and therefore
defined here as reference coverage (Supplementary Fig. 6a). To search
for potential ecDNA fragments, we first divided the whole genome into
bins of 100,000 bp, quantified the read coverage of each bin in each
cell after normalizing the sequencing depth. We then calculated the
relative read coverage as the ratio of the coverage of each bin to the
reference coverage of the cell. Since ecDNAs have much higher
accessibility, regions with a relative coverage of at least six were
defined as potential ecDNA fragments, and further filtered by TSS
enrichment score and cell number (more detail in "Methods") (Sup-
plementary Fig. 6b–d). In addition, given the sparsity of scATAC-seq
data, even an ecDNA derived from a continuous genomic region could
be detected as more than one ecDNA fragment based on scATAC-
seq data if the region was not evenly and sufficiently covered by scATAC-
seq reads due to experimental and/or mapping biases. Therefore, the
number of the ecDNA fragments that we have identified should be
higher than the number of different ecDNA species found in the cell.
To address this, we further investigate the correlation of their read
coverage across different potential ecDNA fragments in individual cell
line and merged nearby and highly correlated fragments as potential
ecDNA regions (see "Methods"). We then defined regions that have
been merged as potential ecDNA regions.

We applied our method to all the cell lines with scATAC-seq data.
As shown in Supplementary Data 5 and Fig. 6a, a median of 229
potential ecDNA regions was detected in each of the 31 cell lines. The
length distribution of potential ecDNA regions ranged from 0.1 M to
30 M with a median of 0.60 M (Supplementary Fig. 6f). Importantly,
our method successfully discovered the experimentally confirmed
ecDNAs, such as chr9:130700000-131400000 in K562, in which *ABL1*
and *NUP214* are located[11]. Oncogenes were significantly enriched in
potential ecDNA regions (Fig. 6a). The result, that, compared to ecD-
NAs without oncogenes, ecDNAs with oncogenes appeared in a higher
proportion of cells within individual cell lines (Fig. 6b), suggested that
the ecDNA containing oncogenes provide cells with a growth advan-
tage and therefore heterogeneity. Given that ecDNA could enhance the
amount of transcribed RNA by increasing DNA copy number, we fur-
ther calculated the correlation between the expression of genes loca-
ted on ecDNAs and the relative coverage number of ecDNAs across
each cell and found that they are partially correlated (Fig. 6c). As an
example, gene *KRT14* and *KRT17*, *LGALS2*, *SERPINE1* and *PCOLCE*, which
located in different ecDNAs in SCC-4 cell line, were specifically
expressed in distinct three subclusters within SCC-4 cell line (Fig. 6d,

Supplementary Data 5 and Supplementary Fig. 6g). Although scRNA-
seq and scATAC-seq data couldn't be matched for SCC-4, the cell
proportion with high expression of these genes, identified by scRNA-
seq data, was highly correlated with the cell proportion containing
ecDNA with these genes, identified by scATAC-seq data (Fig. 6e). This
result implied that the high copy number of ecDNA contribute to the
high expression of genes located in ecDNA. To further evaluate ecD-
NAs' influence on transcriptional heterogeneity, we selected MDA-MB-
231, whose scATAC-seq and scRNA-seq data could be integrated
(Fig. 5b–d left), to reflect the copy number of ecDNA in the UMAP
profiled by scRNA-seq data. As shown in Fig. 6f, Chr12:
5900000_7200000 ecDNA was especially enriched in subcluster 0.
Meanwhile, the genes located on this ecDNA, including *CD9*, *LTBR*,
*PTMS*, *TPI1*, *ENO2*, and *C12orf57*, were also highly expressed in sub-
cluster 0 (Fig. 6g and Supplementary Fig. 6h), and *CD9* is one of the top
marker genes for subcluster 0. These results demonstrated that ecD-
NAs could affect gene expression and therefore cell clustering.
Moreover, a diverse range of TFs was found to be located within
ecDNAs (Supplementary Fig. 6i). We then took the SNB75 cell line as an
example to look into specific TFs located in ecDNA region. As shown in
Fig. 5b, c, by utilizing transcriptome and chromatin accessibility data,
we identified five distinct subclusters (R0-R4) within SNB75. Notably,
the potential ecDNA region (chr22:37439388_38039388) exhibited a
significantly higher copy number specifically in subcluster R3 com-
pared to the other clusters (Supplementary Fig. 6j, k). Gene set varia-
tion analysis (GSVA) revealed that all the 29 genes within the ecDNA
region (chr22:37439388_38039388), including the TF *SOX10*, exhibited
significantly higher expression levels specifically in subcluster R3
(Supplementary Fig. 6l). Importantly, the targets of *SOX10* were also
found to be significantly more activated in subcluster R3 (Supple-
mentary Fig. 6m). These results strongly demonstrated that the pre-
sence of ecDNA could impact the activity of specific transcription
factors and further influence their targets, ultimately contributing to
cellular heterogeneity. Taken together, these results suggest that
ecDNAs could change the copy number of specific genes and therefore
contribute to transcriptomic heterogeneity.

## The transcriptomic heterogeneity dynamics

Recent work has revealed that even single-cell-derived populations of
cancer cells harbor subpopulations, which are marked by fluctuations
of particular gene expressions and could respond differently upon
drug treatment[38,39]. To experimentally explore the relative contribu-
tion of this intrinsic plasticity to the observed transcriptome hetero-
geneity, we chose MDA-MB-231 and HCT 116 cell lines showing discrete
and continuous patterns, respectively, for lineage tracing experiments.
In brief, each cell was first transduced with a lentiviral particle that
encoded a PuroR-T2A-GFP with a unique 28-SW nucleotide barcode
(termed Cell-ID) in its 3'UTR ("Methods", scheme Supplementary
Fig. 7a). We seeded 200 cells carrying Cell-ID for each cell line and
expanded cells for approximately 10 doublings to establish the starting
population and further cultured them for another two weeks. We then
performed scRNA-seq for the two cell lines at two different time points
(T1 and T2). Cells at different time points showed similar clustering
patterns with no significant changes in the proportion composition of
subgroups in both HCT 116 and MDA-MB-231 ($P > 0.05$), demonstrating

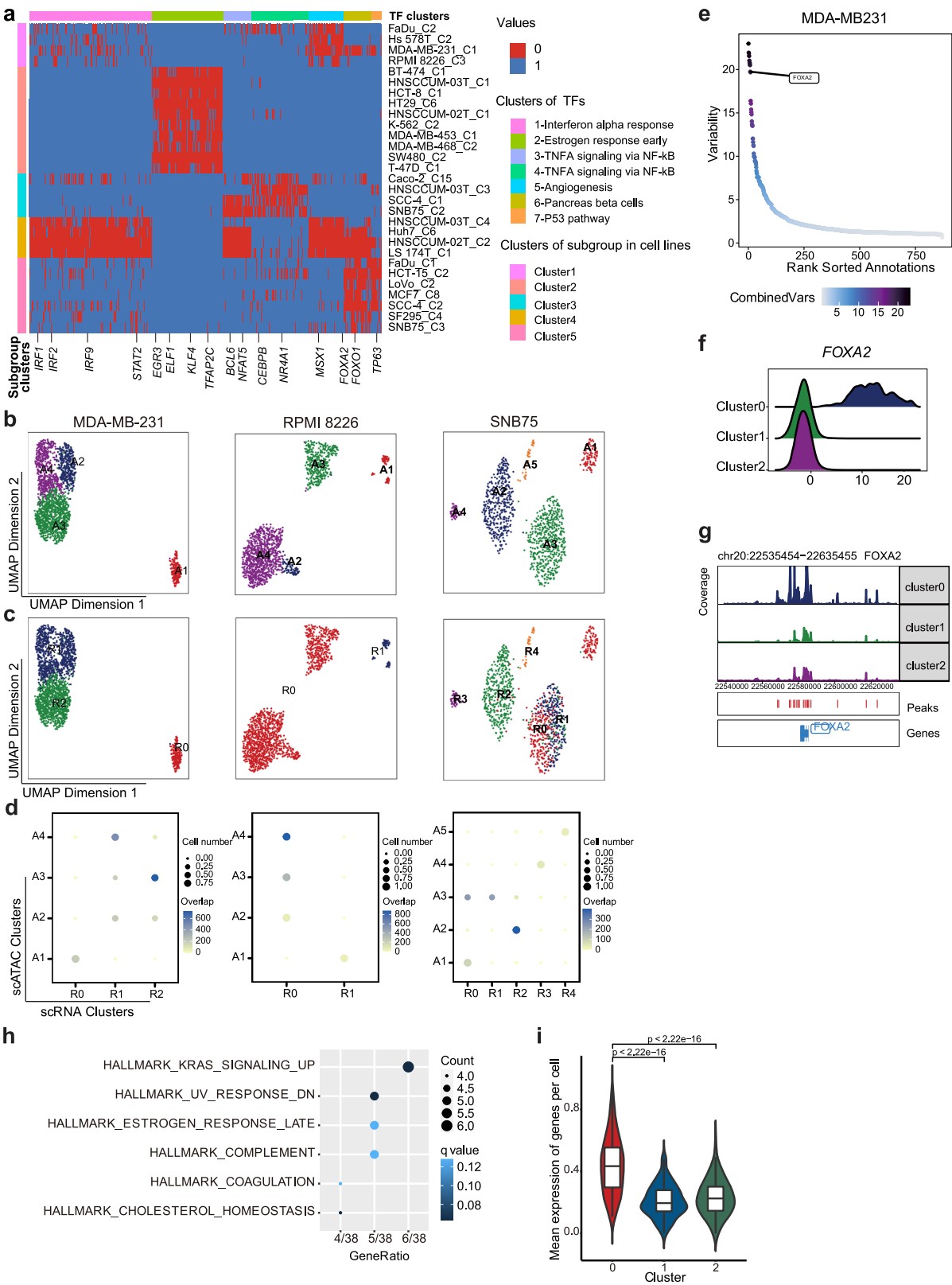

a stable cellular composition along with the cell expansion (Fig. 7a and Supplementary Fig. 7b). Based on the scRNA-seq data, we calculated the barcode number in different time points. At T1, we detected a total of 93 and 87 distinct cell barcodes in HCT 116 and MDA-MB-231, respectively (Fig. 7b and Supplementary Fig. 7c). At T2, we detected a total of 88 and 92 distinct cell barcodes in HCT 116 and MDA-MB-231, respectively (Fig. 7b and Supplementary Fig. 7c). In HCT 116 cells at T1,

only 34 of 93 barcodes were exclusively observed in one subgroup (21 for cluster 0, 4 for cluster 1, and 9 for cluster 2), whereas the cells labeled with the remaining 59 barcodes were distributed in more than one subgroup. Given that the cells labeled with the identical barcode were likely derived from the same ancestor, the observation suggested that one cell could evolve different transcriptomic features during expansion. Among the 34 barcodes, 14 could still be detected in cells at

**Fig. 5 | Expression heterogeneity regulated by epigenetic plasticity. a** The heatmap depicted the hierarchical clustering of cell lines and TFs heterogeneously activated in more than three or at least four cell lines. The exact number and identity of cell lines for each TF are listed in the accompanying source data file. Five clusters of subgroups of cell lines and seven functional groups of TFs were identified, respectively. **b–d** Integration of scRNA-seq and scATAC-seq of three cell lines (MDA-MB-231, RPMI 8226, and SNB75): **b** UMAP plot scATAC-seq clusters of three cell lines; **c** Mapping between transcriptomic subclusters and scATAC-seq clusters. UMAP plot scATAC-seq data with transcriptomic subcluster labels indicated. **d** Dot plot showing the corresponding scRNA-seq clusters (X-axis) and scATAC-seq clusters (Y-axis). Dot color indicates the matching degree between scRNA clusters and scATAC clusters, and dot size indicates the cell proportion in the scATAC-seq cluster that overlaps with corresponding scRNA-seq clusters. A/R combined with the original cluster number was indicated, e.g., A1 represented cluster 1 from

scATAC-seq data, and R0 represented cluster 0 from scRNA-seq data. **e** TFs sorted with the motif enriched in the peaks of heterogenous accessibility of MDA-MB-231. **f** Ridge plot showed the activity of *FOXA2* across subclusters of MDA-MB-231. **g** The chromatin accessibility of *FOXA2* gene locus in different subclusters of MDA-MB-231. **h** Functional enrichment analysis of *FOXA2* downstream target genes with Hallmark gene set. A one-tailed hypergeometric test was used to test the statistically significant differences. FDR-adjusted *p*-value < 0.05. Q value for Kras_signaling_up is 0.074. **i** Comparison of the average expression level of 'KRAS signaling up' related genes among *FOXA2* target genes across subclusters of MDA-MB-231 (*n* = 208 in subcluster 0, *n* = 687 in subcluster 1, and *n* = 724 in subcluster 2). For each boxplot, the center line represents the median, the box indicates the upper and lower quartiles and the whisker represents 1.5-fold of the interquartile range. A two-sided Wilcoxon test was used to test the statistical significance. Source data are provided in the Source Data file.

T2 (Fig. 7b). Among the 14 barcodes, eight barcodes were exclusively distributed in cluster 0 at T1 (appeared in more than one cell), but only two of them were restricted in cluster 0 at T2 and the remaining ones were detected in other subgroups at T2 (Fig. 7c, left panel). For the other six barcodes, which were observed only in cluster 2 at T1, none of them was restricted in cluster 2 at T2 (Fig. 7c, right panel). A similar result was also observed in the MDA-MB-231 cell line (Supplementary Fig. 7b–d). These results demonstrated that although the transcriptome heterogeneity is retained, the transcriptome of individual cells is highly plastic and able to transit between different states, and this phenomenon is common for discrete and continuous cell lines.

### Hypoxia treatment remodels intra-cell-line heterogeneity

Tumor initiation and progression are dynamic processes, which are accompanied by dramatic changes in the microenvironment. Microenvironment plays important roles in tumor progression, therapy resistance, and metastasis formation, and adds another layer of regulation that could drive intratumoral heterogeneity[18]. Here, we sought to evaluate if these transcriptomic heterogeneities are static or rather plastic under different conditions. Given that in solid tumors, tumor cells are often under different extents of hypoxia, hypoxia treatment was chosen as environmental stress for perturbation, and scRNA-seq was performed after 24-h hypoxia incubation on three cell lines (Supplementary Table 5). The result showed that hypoxia-related genes were significantly changed after hypoxic treatment (Supplementary Fig. 8a). To match cell subclusters before and after hypoxia treatment, we utilized Seurat's FindIntegrationAnchors to find anchors between the two datasets, which could reflect cell-to-cell correlations ("Methods"). As a result, we observed two types of correlating pattern: (1) the clusters were one-to-one matched before and after hypoxia treatment, which suggested the difference between cell clusters is not changed by hypoxia treatment, such as ZR-75-1 (Fig. 8a); (2) some clusters were divided from one to more, which indicates that cells within one cluster may respond differently to the hypoxia, such as DLD-1 and SW620 (Fig. 8c, e). Notably, despite the influence of hypoxia, the distinct properties and specific features of different subclusters were maintained across both normal and hypoxia conditions, as evidenced by the significant overlap between the DEGs (differentially expressed genes) identified across clusters under normal and hypoxia conditions (Supplementary Fig. 8b). To identify the potential mechanisms that caused the differential responses to hypoxia stress within one subcluster, we focused on the subclusters that were split under hypoxia. For instance, subcluster 0 of DLD-1 was divided into 2 clusters (cluster 0-1 and 0-2) under the hypoxia condition (Fig. 8c). We examined the differentially expressed genes (DEGs) between cluster 0 and 0-1/0-2: the most variable genes between cluster 0 and 0-1 are hypoxia-related genes such as *BNIP3L*, *SLC2A1* and *ERO1A*, while there are limited hypoxia-related DEGs between cluster 0 and 0-2 (Fig. 8d and Supplementary Fig. 8c). This indicates that there were two groups of cells in the original cluster 0, which responded differently to

the hypoxia, but the difference was not evident at the transcriptional level before hypoxia treatment (Fig. 8d and Supplementary Fig. 8c). Similarly, in SW620 cells, cluster 0 was divided into cluster 0-1 and 0-2 (Fig. 8e); although all cells responded to hypoxia, the cells in cluster 0-1 showed changes of additional genes related to EMT (Fig. 8f).

Previous studies showed that changes in histone modifications and binding of specific TFs may precede and foreshadow changes in gene expression, indicating primed chromatin states influence cell fate by potentiating genes for activation or repression[40,41]. Here, by crosschecking scRNA-seq and scATAC-seq data, we aimed to identify such primed accessible chromatin, which might underlie the different responses to hypoxia treatment. For this purpose, we compared the scATAC-seq data under normoxia with scRNA-seq data under hypoxia and identified cell subclusters that could be confidently matched between the two datasets (Fig. 9a, c). Within DLD-1 cells, subcluster 0-1 was sensitive to hypoxia. scATAC-seq data revealed that ETS (*ELK4*, *FLI1*) and E2F (*E2F3*, *E2F6*) families were activated in subcluster 0-1 cells, which were sensitive to hypoxia. This is in line with the previous studies showing the role of ETS transcription factors in hypoxia-induced gene expression (Fig. 9b)[42,43]. Within SW620 cells, subcluster 0-1 showed changes in EMT-related genes in addition to hypoxia-related ones. Based on scATAC-seq data, we found that HMG (*TCF4*, *TCF3*) family members were the differential TFs in cluster 0-1 (Fig. 9d). Given previous studies showed that HMG family could regulate EMT, this implies that these TFs might contribute to the hypoxia-induced EMT activation[44,45]. These results suggested that heterogeneous chromatin accessibility could contribute to the heterogeneous stress response of a seemingly homogenous cell population.

## Discussion

In this study, we provided a multi-omics single-cell profiling of dozens of human cancer cell lines of nine tissue origins. Detailed analysis revealed that transcriptomic heterogeneity was frequently observed in established cancer cell lines and shaped by multiple common transcriptional programs. CNV inferred by scRNA-seq, epigenomic divergence, and extrachromosomal DNA (ecDNA) distribution measured by scATAC-seq were found to contribute to the observed intra-cell-line heterogeneity. Through lineage tracing and hypoxia treatment, we found that transcriptomic heterogeneity is dynamic and could be reshaped by environmental stress.

The intra-cell-line heterogeneity observed in commonly used cancer cell lines may have a profound influence on cell line-based biomedical research. It may change the way researchers interpret their results and provide new explanations for unsolved questions. Our previous study has shown that cellular response to PARP inhibitors could be affected by the EMT program, which was heterogeneously activated within the cell line[5]. Srivatsan et al. also found heterogeneity in the cellular response to HDAC inhibitors, which is caused by the variation in cells' acetate reservoirs[46]. The transcriptional variation revealed by scRNA-seq could provide information on how

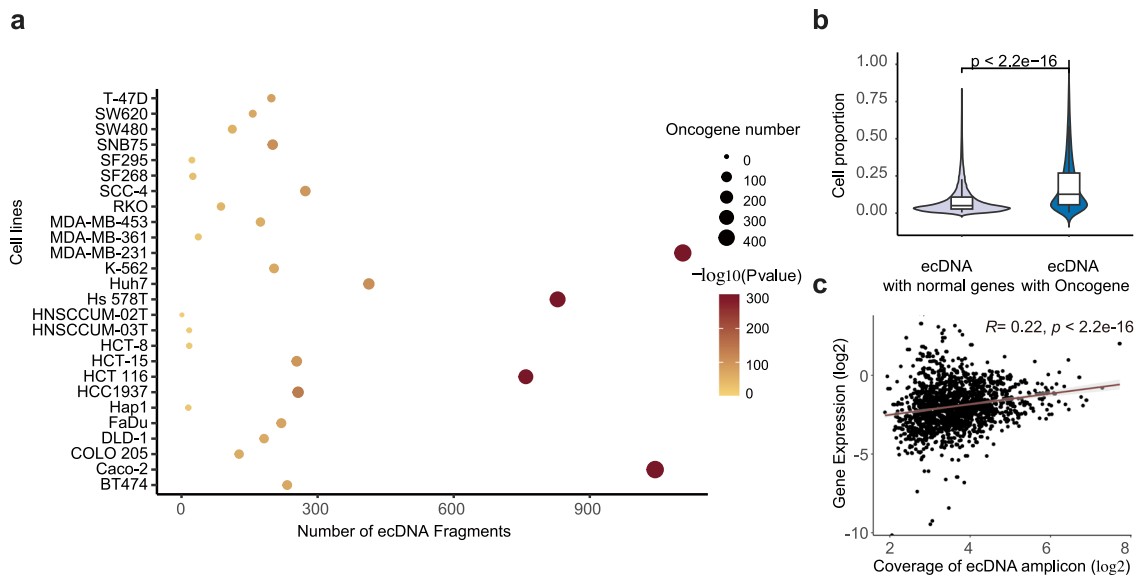

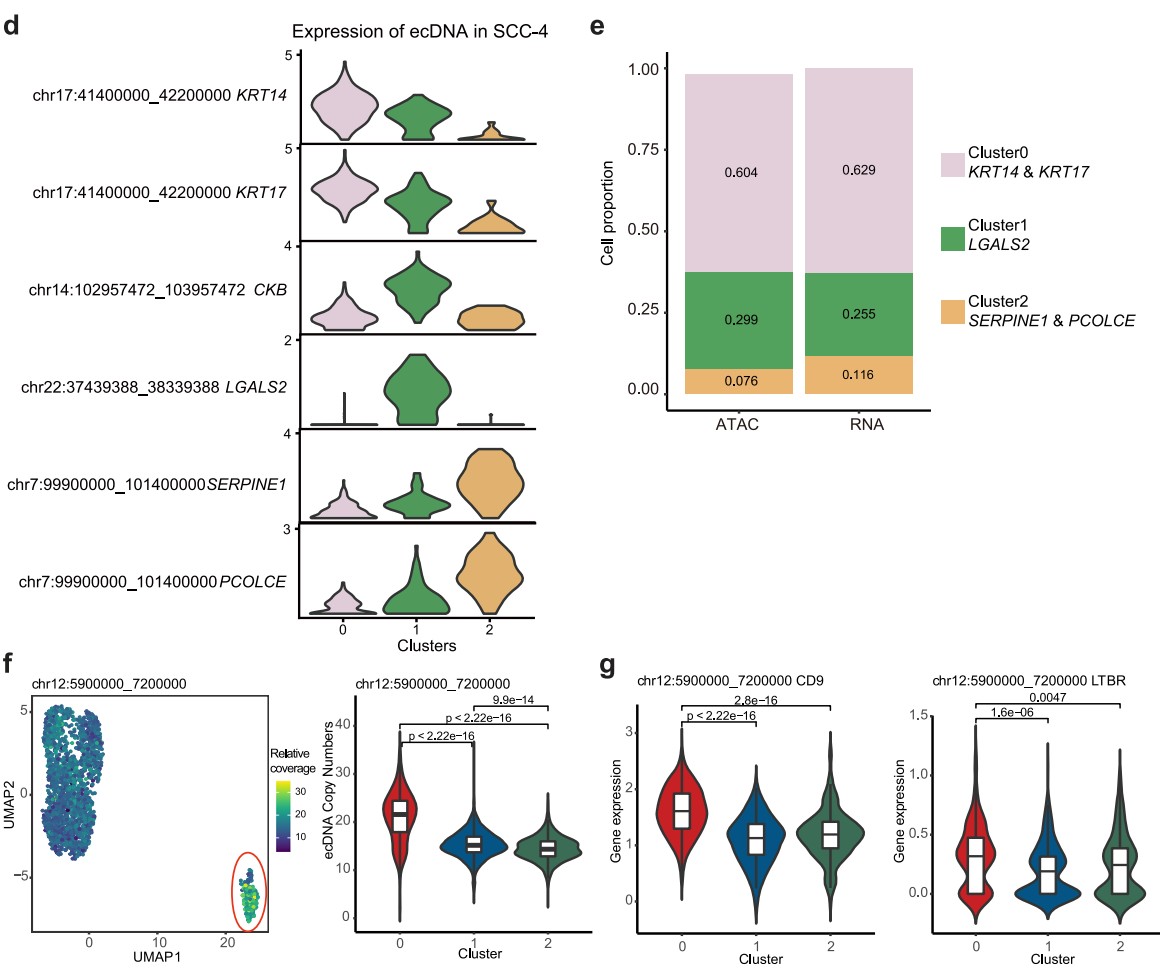

heterogeneously cells may react to drug treatment and how mechanistically drug resistance may occur. Moreover, scRNA-seq data provide more precise information for molecular subtype classification of the cancer cell lines, which may help to identify cell lines resembling specific features of human tumors. Meanwhile, scATAC-seq data reveal cellular heterogeneity at a different regulatory level. As a resource, it offers a comprehensive dataset for the analysis of intra-cell-line

heterogeneity at both the transcriptomic and epigenomic levels, which could be used as a reference to choose suitable cell lines for intended studies, such as mechanistic investigation, drug tests, and high-throughput screening.

Besides illustrating the intra-cell-line heterogeneity at a given time, our study also measured the dynamic nature of cellular states by combining the cellular genetic barcode tracing technique and

**Fig. 6 | Contribution of ecDNAs to the cellular transcriptomic heterogeneity. a** The number of ecDNA in different cell lines and ecDNA amplicons with oncogenes accumulate in cells. A one-tailed hypergeometric test was used to test the statistical significance. **b** ecDNAs with oncogenes (*n* = 2149), compared to ecDNAs with non-oncogenes (*n* = 5469), appeared in a higher proportion of cells within individual cell line. For each boxplot, the center line represents the median, the box indicates the upper and lower quartiles and the whisker represents 1.5-fold of the interquartile range. Each dot stands for the cell proportion of cells with one specific ecDNA within an individual cell line. A two-sided Wilcoxon test was used to test the statistical significance. **c** The correlation between the relative coverage number of ecDNAs and the RNA expression level of genes that appeared in ecDNAs. Spearman's rank correlation coefficient was used to evaluate the correlation between the scATAC-seq read coverage of ecDNA amplicon (*x*-axis) and the RNA expression level. A two-tailed Spearman correlation test was used to test statistical significance. **d** The expression of genes appearing in ecDNA regions in the subcluster of SCC-4.

**e** The percentage of ecDNA-positive cells and high expression cells was correlated in SCC-4. **f** UMAP map of coverage of ecDNA (chr12: 5900000_7200000). Each dot represents a cell, the color from blue to yellow represents the coverage from low to high, and the red circle marked out is cluster 0 of MDA-MB-231 (*n* = 208 in cluster 0, *n* = 687 in cluster 1, and *n* = 724 in cluster2). For each boxplot, the center line represents the median, the box indicates the upper and lower quartiles and the whisker represents 1.5-fold of the interquartile range. A two-sided Wilcoxon test was used to test the statistical significance. **g** The expression of genes located on ecDNA (chr12: 5900000_7200000) in different clusters (*n* = 208 in cluster 0, *n* = 687 in cluster 1, and *n* = 724 in cluster2). For each boxplot, the center line represents the median, the box indicates the upper and lower quartiles and the whisker represents 1.5-fold of the interquartile range. A two-sided Wilcoxon test was used to test the statistical significance. Source data are provided in the Source Data file.

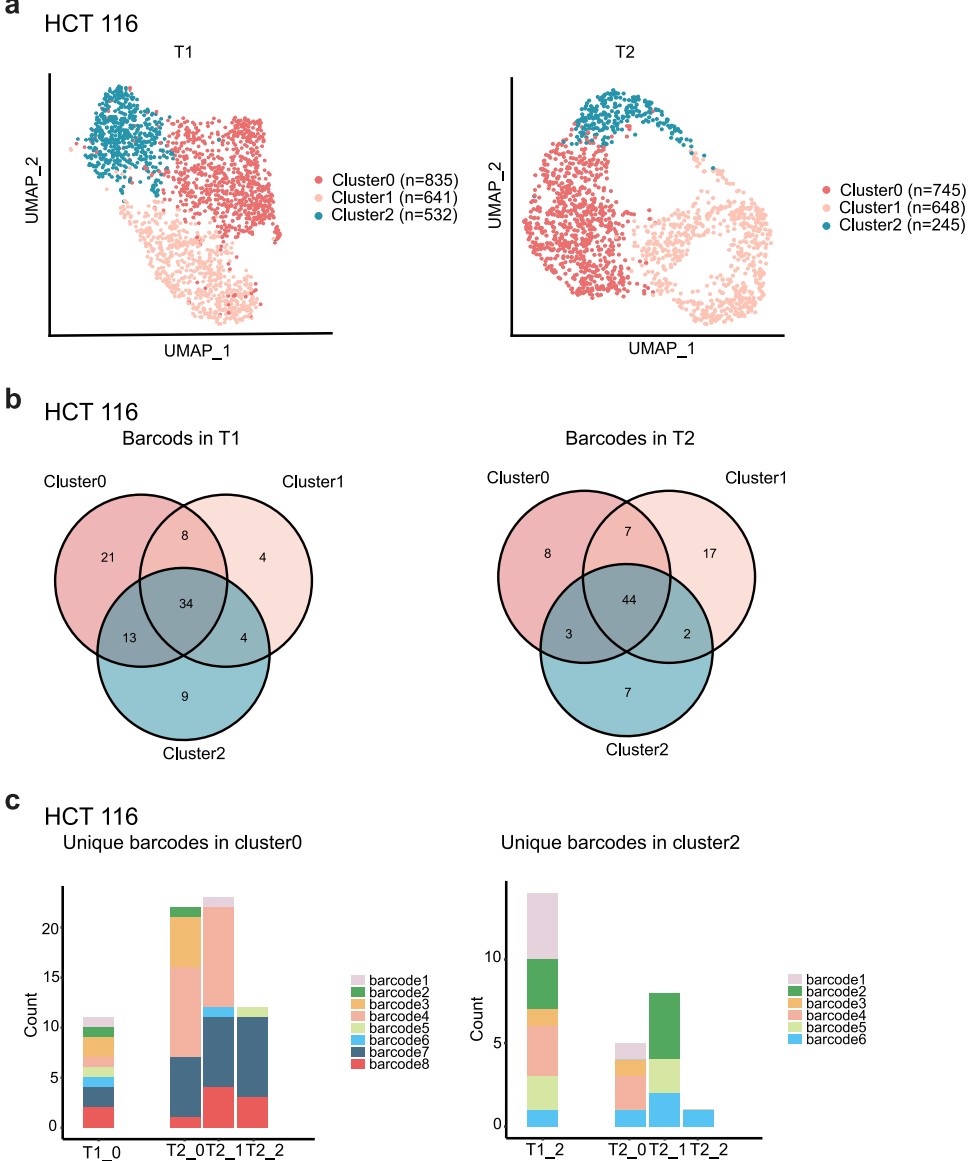

**Fig. 7 | The plasticity of cellular transcriptomic heterogeneity. a** A UMAP plot of HCT 116 at time point 1 (T1, left) and time point 2 (T2, right) (*n* = 2008 cells at T1, *n* = 1638 cells at T2). *N* represents cell numbers in different subclusters. **b** Venn diagram of barcodes in different subclusters of HCT116 at T1 (left) and T2 (right).

**c** The distribution of unique barcodes in subclusters at T1 was rearranged at T2 for HCT 116. Time point combined with the original cluster number was indicated, e.g., T1_0 represented cluster0 from T1. Source data are provided in the Source Data file.

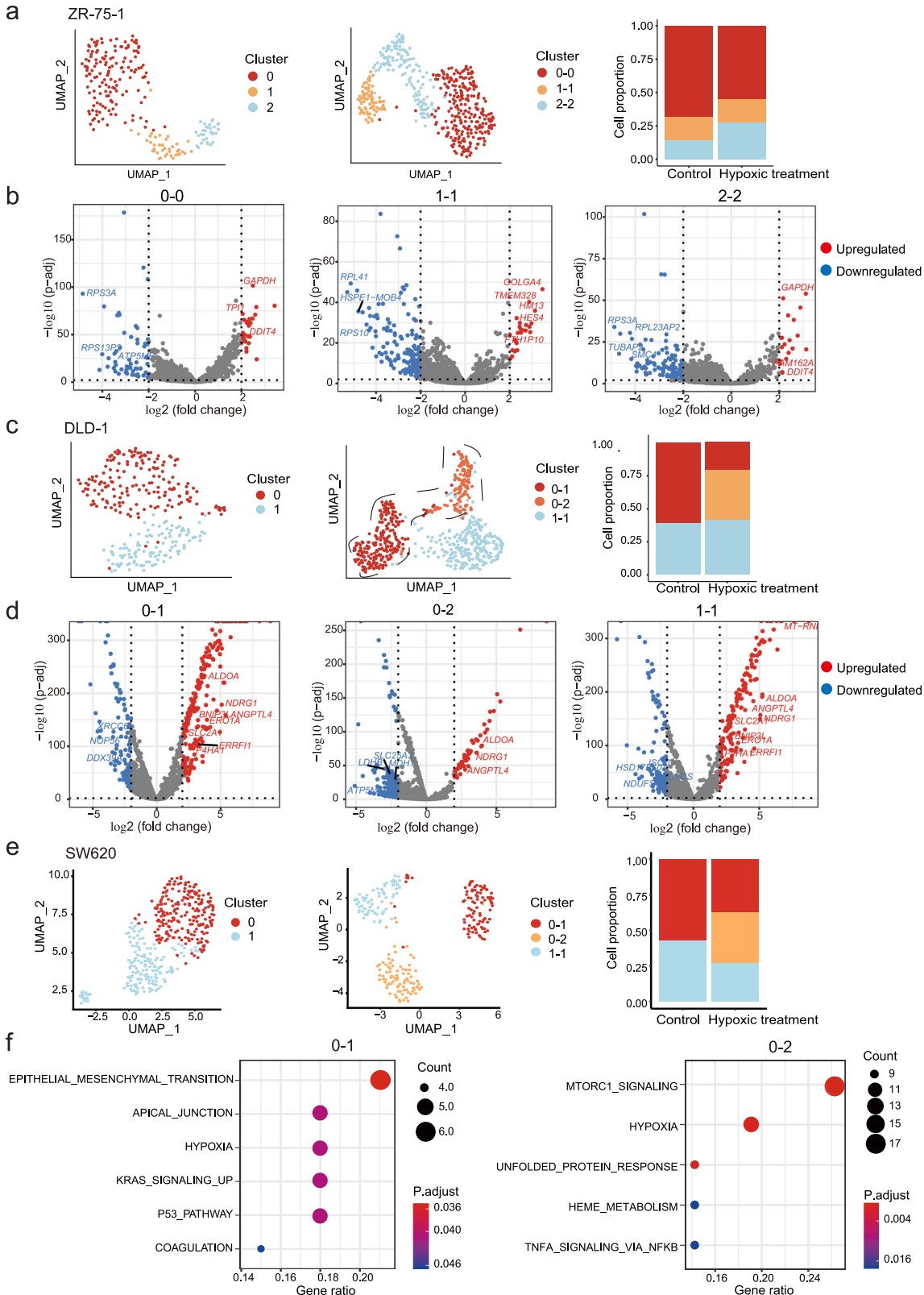

demonstrated that, in some cell lines, the daughter cells from the same progenitor might enter different transcriptomic statuses, suggesting that even apparently homogeneous cells may have latent variability. Such latent variability was also observed in the hypoxia experiment, where hypoxia treatment drove the heterogeneous cellular response from the same cell line, even from the same subpopulation. the emergence of a new transcriptomic subcluster in DLD-1 and SW620

cell lines. This suggests that transcriptomic heterogeneity within cancer cell lines may be far more complex than we observed under normal culture conditions, or the current scRNA-seq technique is not sensitive enough to reveal the latent variability. Therefore, we further investigated if the latent transcriptomic variability could be observed at the epigenomic level by using scATAC-seq, given epigenomic heterogeneity may reflect not only the present but also the past and future

**Fig. 8 | Cells responded differently under hypoxia. a** The subclusters could be matched before and after hypoxia treatment in ZR-75-1. Left, UMAP plots scRNA-seq of ZR-75-1 before hypoxia treatment. Middle, UMAP plots scRNA-seq of ZR-75-1 upon hypoxia treatment. Right, the barplot shows the abundance of clusters in ZR-75-1 with or without hypoxia treatment. Different clusters are labeled with different colors. **b** Volcano plot showing log2 fold change (x-axis) and −log10-transformed FDR-adjusted P-value (y-axis) of differentially expressed genes in corresponding clusters of ZR-75-1 between control and hypoxia treatment (n = 83 DEGs in corresponding cluster 0 of ZR-75-1 between control and hypoxia, n = 182 DEGs in corresponding cluster 1 of ZR-75-1 between control and hypoxia, n = 144 DEGs in corresponding cluster 2 of ZR-75-1 between control and hypoxia). A moderated t-statistics and empirical Bayes methods was used to test statistical significance. Upregulated genes are shown in red, and downregulated genes are highlighted in blue. Insignificant genes are colored in gray. **c** Cells in the same subcluster responded differently to hypoxia in DLD-1. Left, UMAP plots scRNA-seq of DLD-1 before hypoxia treatment. Middle, UMAP plots scRNA-seq of DLD-1 upon hypoxia treatment. Right, the barplot shows the abundance of clusters in DLD-1 with or without hypoxia treatment. Different clusters are labeled with different colors.

**d** Volcano plot showing log2 fold change (x-axis) and −log10-transformed FDR-adjusted P-value (y-axis) of differentially expressed genes in corresponding clusters of DLD-1 between control and hypoxia treatment (n = 337 DEGs between cluster 0 and 0-1, n = 422 DEGs between cluster 0 and 0-2, n = 332 DEGs in corresponding cluster 1 of DLD-1 between control and hypoxia). A moderated t-statistics and empirical Bayes methods was used to test statistical significance. Upregulated genes are shown in red, and downregulated genes are highlighted in blue. Insignificant genes are colored in gray. **e** Cells in one subcluster responded differently to hypoxia in SW620. Left, UMAP plots scRNA-seq of SW620 before hypoxia treatment. Middle, UMAP plots scRNA-seq of SW620 upon hypoxia treatment. Right, the barplot shows the abundance of clusters in SW620 with or without hypoxia treatment. Different clusters are labeled with different colors. **f** Functional enrichment analysis of differentially expressed genes in subclusters of SW620 between control and hypoxia treatment (n = 93 DEGs between 0 and 0-1, n = 190 DEGs between 0 and 0-2). A one-tailed hypergeometric test was used to test the statistical significance. EMT-related genes were differentially induced by hypoxia treatment in one subcluster of SW620. Source data are provided in the Source Data file.

---

cell status. Indeed, within one transcriptomic subcluster, differential primed chromatin states driven by specific TFs preceded hypoxia. This reminds us that when targeting observed heterogeneous transcriptional programs for cancer therapy, cells may evolve new mechanisms to overcome the treatment and the knowledge of epigenetic heterogeneity may help us to predict and prevent it from happening.

Furthermore, in this study, scATAC-seq was utilized for the first time to characterize the ecDNAs. Our data revealed the pervasive presence of ecDNA in cancer cell lines. The ecDNAs could drive high gene expression of the contained genes and their variable copy numbers in individual cells contribute to the observed transcriptional heterogeneity. Compared to previously published techniques, including AmpliconArchitect[47], Circle-seq[48], and Circle-Finder[49] that are based on bulk sequencing, our method developed for analyzing single-cell ATAC-seq data could identify highly potential ecDNA regions at single-cell resolution and therefore reveal specific events in rare cell populations, which could be neglected as noise in bulk sequencing data[47–49]. We observed on average hundreds of ecDNA events in individual cell lines, including ecDNAs that were validated by FISH and other sequencing-based techniques[11]. The ecDNAs identified by our method potentially contained a higher fraction of false positives than bulk sequencing-based approaches, owing to the fact that with the limited scATAC-seq read length, we could not use junction site reads for further filtering as in those methods. Nevertheless, the results derived from our single-cell data could provide important resources for further validation experiments using imaging-based tools.

In conclusion, this study advances the understanding of cellular heterogeneity within commonly used cancer cell lines and provides insights into the molecular mechanism driving it. We envision that the data and analysis provided in this study will facilitate all sorts of cancer cell line-based biomedical research, form a basis for investigating intratumor heterogeneity using available cancer cell lines, enable new biological insights into different research models, and eventually contribute valuable insights into tumor biology and have implication for the further development of precision therapies.

## Methods
### Cell line cultures
The 786-O (CRL-1932), A-253 (HTB-41), A549 (CCL-185), BT-474 (HTB-20), BT-549 (HTB-122), Caco-2 (HTB-37), COLO 205 (CCL-222), DLD-1 (CCL-221), FaDu (HTB-43), HCC1937 (CRL-2336), HCT 116 (CCL-247), HCT-15 (CCL-225), HCT-8 (CCL-244), HeLa (CCL-2), Hep G2 (HB-8065), HK-2 (CRL-2190), Hs 578T (HTB-126), HT-29 (HTB-29), K-562 (CCL-243), LoVo (CCL-229), LS 174T (CL-188), MCF7 (HTB-22), MDA-MB-231 (HTB-26), MDA-MB-361 (HTB-27), MDA-MB-453 (HTB-131), MDA-MB-

468 (HTB-132), RKO (CRL-2577), RPE-1 (CRL-4000), RPMI 8226 (CCL-155), SCC4 (CRL-1624), SK-BR-3 (HTB-30), SW480 (CCL-228), SW620 (CCL-227), T-47D (HTB-133) and ZR-75-1 (CRL-1500) cells were obtained from the ATCC. The HAP1 (C631) cells are from Horizon Discovery and Huh7 (SCSP-526) cells are from The Cell Bank of the Chinese Academy of Sciences. The SF268 and SNB75 cells are from NCI, and the SF295 (C0005005) cells are from AddexBio. The HUNSCCUM-02T and HUNSCCUM-03T cells were kindly gifted by Walter Birchmeier's lab at Max-Delbrueck-Center for Molecular Medicine, Berlin. All cell lines used in this study were maintained in DMEM (Gibico, #11995040) or RPMI1640 (Gibico, #22400089) medium supplemented with 10% fetal bovine serum (FBS, Gibico, #10270106), 1% Penicillin/Streptomycin (P/S, Gibico, #15070063) in a humidified atmosphere of 5% $CO_2$ at 37 °C. Cell line identity was confirmed by the Multiplex human cell line Authentication Test (MCA, Mutiplexion) or Cell Line Authentication (GENEWIZ, Suzhou). We regularly checked all cell lines for mycoplasma contamination.

### Hypoxia treatment
The normoxic set of plates was placed in an aerobic incubator. To create hypoxia, the hypoxic set was moved to a Hypoxia Incubator Chamber (StemCell) which was equilibrated at 37 °C in a humidified atmosphere of 5% $CO_2$ and 1% $O_2$. After 24 h incubation, the cells, which were 80–90% confluent, were washed once by PBS, and then harvested with 0.05% Trypsin-EDTA. After neutralization with complete medium, centrifugation, and resuspension in PBS with 0.04% BSA, the cells were measured.

### Cloning procedures
The vector encoding PuroR-T2A-GFP protein followed by a 28-SW sequence in its 3'UTR was constructed based on CROPseq-Guide-Puro (Addgene#86708) by inserting a T2A-GFP coding sequence after PuroR and replacing the U6 promoter and filler sequences with a 28-SW sequence.

### Virus package and transduction
HEK293T cells were seeded one day before and were transfected with lentiviral plasmid, psPAX2 (Addgene), and pMD2.G (Addgene) at ratio of 4:3:1 using PEI (Sigma). The medium was replaced 12 h after transfection. The supernatant was collected after 48 h with centrifugation and filtering through a 0.45 μm filter. The transduction was worked by incubating the viral particles containing supernatant with the targeting cells overnight in the presence of polybrene (Sigma). To ensure each cell was labeled with only one distinct barcode, we transfected 150,000 cells with a barcode library consisting of 230,000 distinct barcodes at a MOI of 0.1.

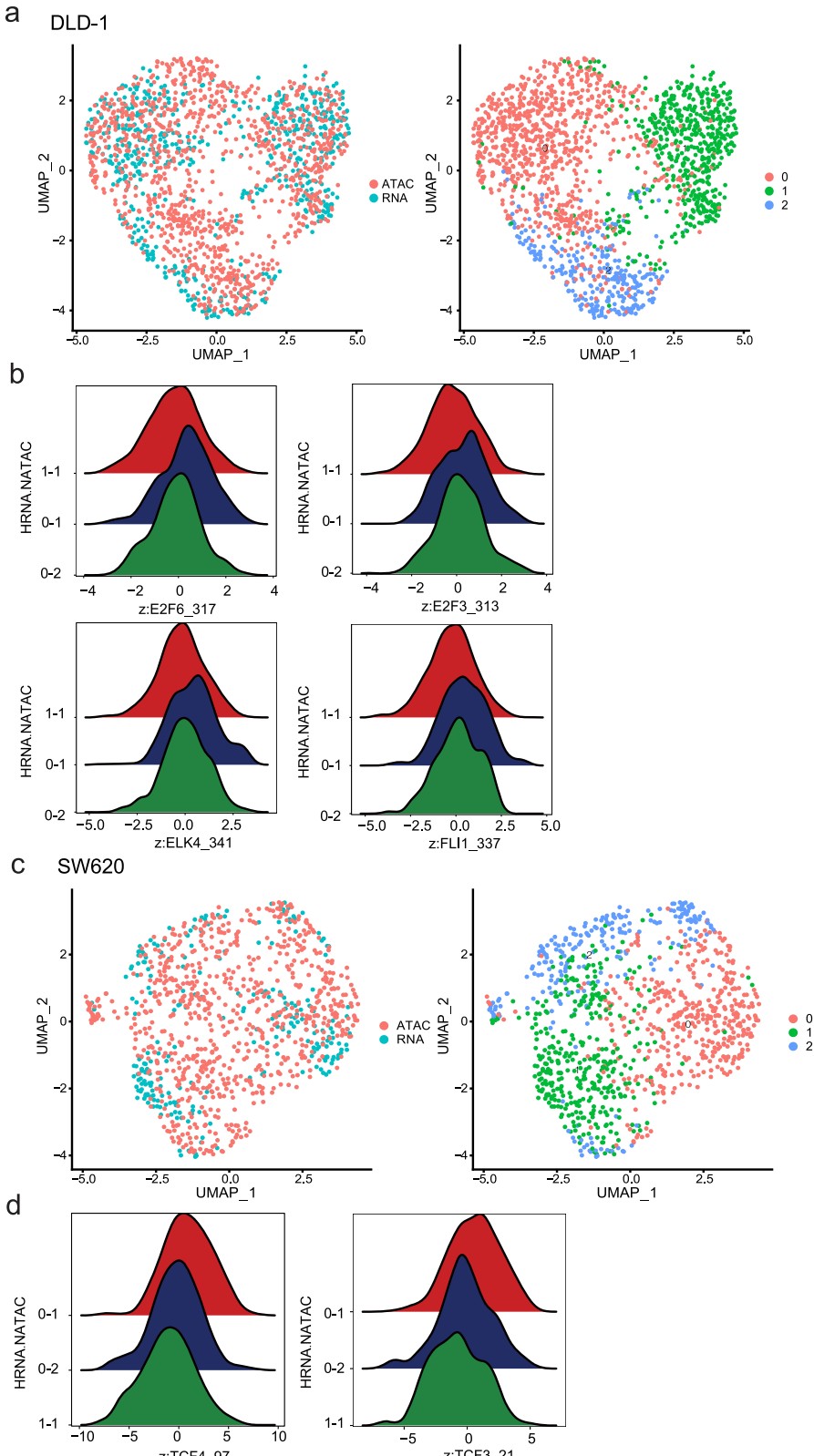

**Fig. 9 | TFs are responsible for heterogeneous response under hypoxia treatment.** **a** UMAP plot of integration of scATAC-seq under normoxia and scRNA-seq with hypoxia treatment of DLD-1 (left). UMAP plot of scRNA-seq of DLD-1 upon hypoxia treatment (right). **b** The activity of ETS (*ELK4*, *FLI1*) and E2F (*E2F3*, *E2F6*) family TFs across clusters in DLD-1. **c** UMAP plot of integration of scATAC-seq under normoxia and scRNA-seq with hypoxia treatment of SW620 (left). UMAP plot of scRNA-seq of SW620 upon hypoxia treatment (right). **d** The activity of HMG (*TCF4*, *TCF3*) family TFs across clusters in SW620. Source data are provided in the Source Data file.

## scRNA-seq with DNBelab C4 system

Single-cell RNA-seq of all cell lines, with and without hypoxia treatment, was performed with the DNBelab C4 Series Single-Cell Library Prep Set (MGI, #1000021082) according to the corresponding protocols[50,51]. Cells were harvested with 0.05% Trypsin-EDTA. After neutralization with complete medium, centrifugation, and resuspension in PBS with 0.04% BSA, the cells were filtered through a 35-μm Cell Strainer (Corning 352235) and a single-cell suspension was prepared. Cells and barcoded beads were diluted at the concentration of 1000 cells/μL in cell resuspension buffer and 1000 beads/μL in lysis buffer, respectively. For each pool containing two or three cell lines in equal quantity, 100,000 single cells and an equal number of beads were used for droplet generation. After incubation at room temperature, the droplets were broken, and the beads were recovered, reverse transcribed, and amplified to generate sufficient cDNA.

## scATAC-seq sample preparation

For all cell lines, single-cell ATAC-seq libraries were generated by DNBelab C Series Single-Cell ATAC Library Prep Set (MGI, #1000021878). Single cells (80% viability) were lysed and transposed to generate single nucleus suspensions according to the corresponding protocols[50]. Successful preparation of intact, isolated nuclei was confirmed through visual inspection in phase-contrast microscopy, and nucleus concentration was assessed before proceeding immediately to transposing with transposase (BGI, China). For each pool, 10,000 single nuclei containing four to six cell lines in equal quantity and an equal number of beads were used for droplet encapsulation. Through further procedures including pre-amplification, emulsion breakage, capture beads collection, DNA amplification, and purification, dsDNA fragments were generated.

## scRNA-seq library preparation and sequencing

Indexed sequencing libraries were prepared according to the operating standards for MGI library preparation. The quality control of ssDNA libraries was performed using QubitTM ssDNA Assay Kit and DNA nanoballs (DNBs) were sequenced on BGISEQ500 sequencer or DIPSEQ T1 sequencer.

## Processing of scRNA-seq data

Raw sequencing reads were processed using the PISA (v0.4). The reads were aligned to the hg38 reference genome by STAR (v.2.7.9). Aligned reads were then used to parse out the cell barcodes and unique molecular identifier (UMI). Next, PISA corr was used to correct the UMI or barcode sequence based on Hamming distance. UMI counting defined by PISA was further used with default parameters to generate a cell versus gene UMI count matrix. Cells with more than 1000 UMI and 500 genes were retained for further analysis.

## Cell clustering

The Seurat (v.4.1.0) was applied to the expression matrix to perform quality control, normalization and scaling, detection of highly variable features, dimensional reduction and clustering: (1) data were normalized by the LogNormalize function with default parameters; (2) the top 2000 most variable genes of each sample were detected by FindVariableFeatures with the vst method; (3) dimensionality reduction of PCA was performed using RunPCA; (4) to cluster the cells, a KNN graph was constructed based on the Euclidean distance and modularity optimization techniques were applied using FindNeighbors and FindClusters (resolution = 0.3/0.15); (5) the top 15 PCs were used to perform UMAP to visualize the cells; (6) the differentially expressed feature genes across different clusters were detected by the FindAllMarkers function. The cluster biomarkers were defined as genes with adjusted $P$-value < 0.01 and $log_2FC$ > 0.5.

In the process of setting up a UMAP profiling for all cell lines, filtered data in each cell line were used and the procedures of merged data were similar to those for a single cell line, except that we performed FindClusters with resolution 0.5 and RunUMAP with top 30 PCs on merged data.

## Cell line assignment

We used both expression-based and CNA-based methods to assign cells to cell lines. For the expression-based method, marker genes of unidentified cell lines were detected by the FindAllMarkers function in Seurat with adjusted $P$ < 0.01 and $log_2FC$ > 0.5. We then compared these marker genes with the highly expressed genes in external bulk RNA-seq data of the corresponding cell line from Harmonizome[52] to assign the cells to the most similar cell line. For CNV-based methods, InferCNV (v1.8.1) was used for constructing the CNV profile in each cell line. We then compared the CNVs with the reported main CNVs in each cell line from cBioPortal[53]. To further confirm cell line identification, we also compared the single-cell expression data to bulk-seq data from the CCLE database or GEO (SK-BR-3: GSE7562, https://www.ncbi.nlm.nih.gov/geo/query/acc.cgi?acc=GSE7562; LS 174T: GSE18560, https://www.ncbi.nlm.nih.gov/geo/query/acc.cgi?acc=GSE18560)[13]. Only cell lines whose assignments were consistent among different methods were retained for further analysis.

To validate the effectiveness of our assignment, we integrated the bulk RNA-seq dataset generated by CCLE of three cell lines[13], including A-253, HCT 116 and MCF7, with our scRNA-seq data. The analysis showed that pseudo-single cells deriving from the bulk RNA-seq dataset were distributed within the corresponding cell populations on UMAP, and differentially expressed genes across the cell lines were also consistently expressed in both bulk and single-cell datasets (Supplementary Fig. 1c–e).

## Systematic characterization of transcriptional heterogeneity

We analyzed each cell line separately using NMF to identify programs of expression heterogeneity as described before:[8] (1) Data preprocessing was applied by normalization and transforming all negative values to zero. (2) NMF was performed with the number of factor $k$ ranging from 6 to 9, and the top 50 genes were defined as feature genes for the NMF programs. (3) Robust NMF programs that were identified with different factor $k$ were retained. (4) Common NMF programs across cell lines (>3 cell lines) were the focus of further analysis. (5) ClusterProfiler (v.4.2.2) was applied to identify the functional characteristics for each retained common program with adjusted $p$-value (<0.01) and the maximum matching counts.

## Function enrichment

Function enrichment analysis was performed using the enricher function in clusterProfiler (v4.2.2). Only functions with $p$-value < 0.05 were retained.

## Defining diversity scores in each cell line

To quantify intratumoral heterogeneity, we calculate the diversity score of each cell line based on the gene expression profiles[22]. First, we employed PCA to project the expression profiles of all cells to derive PCs, which could capture key features and reduce noise. The top 30 PCs were selected. We then define diversity as the average distance of all cells within the cell line to the centroid. To exclude the impact of outliers, we removed the extreme values that were beyond the range of 3 standard deviations from the mean.

## Cell cycle analysis

We calculated the program cluster score of individual cells using the AddModuleScore function of the Seurat R package with the default parameter. The input data was normalized scRNA-seq data of all cell lines. Then cells with the top 5% program cluster score were selected as representative cells for each program cluster. We assessed the cell

cycle phase of representative cells using the Seurat CellCycleScoring function of the Seurat R package with default parameter.

## Processing of scATAC-seq data

PISA was used to parse out the cell barcodes and UMIs based on the raw FASTQ files of scATAC-seq data. Clean reads were aligned to the hg38 reference genome by BWA (v.2.2)[54]. Deconvolution was performed to remove doublets in scATAC sequencing data. MACS2 (v2.2.7.1) was applied to call peaks. The ArchR (v.1.0.2) was applied to perform quality control, dimensionality reduction, clustering, calculation of gene scores and motif enrichment: (1) those with a low number of TSS proportion (<4) and a low number of unique fragments (<1000) were filtered out; (2) the doublet score for each cell was calculated by the addDoubletScores function and the predicted doublet cells were filtered according to the filterDoublets function with the default parameters; (3) due to the sparsity of the data, Latent Semantic Indexing (LSI) was first used for dimensionality reduction, then singular value decomposition (SVD) was applied to identify the most valuable information across samples in a lower dimensional space; (4) the addClusters function was used for clustering (resolution = 0.3); (5) the addGeneScoreMat function was performed to add gene score for each gene based on the weighted distance from each peak (start or end) to the gene body. The getMarkerFeatures function was used to mark feature identification: each cell group was compared to its own background group to determine if the given cell group had significantly higher accessibility. The plotEmbedding function was applied for visualization.

## Defining diversity scores in each cell line for scATAC-seq

To quantify intratumoral heterogeneity in scATAC-seq, we calculate the diversity score of each cell line based on chromatin accessibility. First, we employed Iterative Latent Semantic Indexing (LSI)[28] to project the chromatin accessibility profiles of all cells to derive LSIs, which could capture key features and reduce noise, for the sparsity of the data. The top 30 LSIs were selected. We then define diversity as the average distance of all cells within the cell line to the centroid. To exclude the impact of extreme values on diversity score calculation, we used mean ± 3*standard deviation to detect extreme values and removed the extreme value-related cells for diversity calculation.

## Transcription factor motif enrichment analysis

To identify key TFs responsible for subgroup-specific accessible chromatin regions, two approaches, including ArchR (v.1.0.2)[28] and ChromVAR (v.1.16.0)[30], were used for identifying TFs with motif enriched in differential ATAC peaks from subgroups of a same cell line. ArchR was employed to identify potential TFs that drive differential chromatin accessibility across cell subclusters. More specifically, for ArchR, addMotifAnnotations was used to determine the motif presented in the peak set and peakAnnoEnrichment was used to identify the motifs enriched in differentially accessible peaks. Based on the motif determined by addMotifAnnotations, ChromVAR was applied to predict the enrichment of TF activity on a per-cell basis. Initially, the chromVAR::getBackgroundPeaks function was used to get background peaks based on similarity in GC-content and the number of fragments across all samples using the Mahalanobis distance, then the addDeviationsMatrix function was used to compute per-cell deviations across all of our annotated motifs.

## TFs clustering

We obtained cluster-specific TF of each cell line which was enriched from differential peak using the ArchR package. Then, we generated a cluster by TF matrix, which only consists of 0 and 1 according to whether the TF existed in the cluster. After hierarchical clustering, five modules were identified. For each module, we selected TFs that existed in over 50% of clusters as main TFs. Finally, these main TFs were

used to perform functional enrichment analysis by clusterProfiler package using the Hallmark gene set.

## Integration of scATAC-seq data with scRNA-seq data

The FindTransferAnchors function implemented in ArchR was used to integrate scATAC-seq and scRNA-seq data by comparing the scATAC-seq gene score matrix with the scRNA-seq gene expression matrix. The anchors were used to transfer cluster-label identifiers between the two data types using the TransferData function. Each cell in the query was assigned the cluster label with the highest prediction score, and only query cells with prediction scores above 0.9 were considered to have been successfully label transferred. Each cell in the scATAC-seq data was aligned to the most similar cell from the scRNA-seq data.

## TFs targets analysis

The SCENIC[55] or pySCENIC (v.0.12.0)[56] workflow was applied to scRNA-seq data of individual cell lines to infer TFs and their target genes. First, GENIE3 (GRNboost) was used to identify potential targets for each TF based on the co-expression regulatory network. Then, RcisTarget identified direct-binding targets (regulons) in gene networks based on motif enrichment analysis. Furthermore, AUCell was applied to determine the AUC score for these regulons in each cell by calculating the enrichment of the regulon as an area under the recovery curve (AUC) across the ranking of all genes in a particular cell, whereby genes are ranked by their expression value.

## GSVA analysis

Gene set variation analysis (GSVA, v1.44.2) was used to estimate the activity of a set of genes by transforming an input gene-by-sample expression data matrix into a corresponding gene-set-by-sample expression data matrix. The average normalized expression for cells was obtained. Then, GSVA scores of gene sets associated with ecDNA or hypoxia were calculated.

## CNV estimation

The copy number variation of single cells was calculated with InferCNV, which analyzes gene expression intensity across the genome based on scRNA_seq data. RPE-1 and HK2 cells were selected as reference normal cells. The inferCNV analysis was performed with parameters including "denoise", default hidden Markov model (HMM) settings, and a value of 0.1 for "cutOff". A heatmap was generated to illustrate the relative expression intensities across each chromosome.

## Association between the CNV subclones and transcriptional subclusters

Hierarchical clustering was employed to detect CNV subclones within individual cell lines. We concentrated on chromosomal locations that differed considerably and reasoned that the presence of multiple CNV subclones in a single cell line would be reflected in a multimodal distribution of CNV signal for at least one chromosome arm across cells. We then classified cell lines into three types (A, B, C). Within the A-type cell line, cells in the same CNV subclone were only present in one transcriptomic subcluster. Within the B-type cell line, cells in the same CNV subclone were scattered across multiple transcriptomic subclusters. C-type cell lines have no CNV subclones.

To further investigate the contribution of CNV to transcriptional variability, we focused on A-type cell lines and extracted genes located within CNV regions. Then the hypergeometric test was used to assess whether these CNV genes overlapped significantly with differentially expressed genes between subclusters.

## EcDNA detection

EcDNA amplicon in single cells was detected based on scATAC-seq data by the following steps: (1) The whole genome was divided into

bins of length 100,000 bp. Then, we calculated the 'normalized coverage' of each bin in each cell as (read counts in each bin) / (fragment counts in the whole chromosome) × 10,000. Given that the coverage distribution shows multiple peaks in every single cell, we defined the first peak as 'reference coverage' for the regions of two genomic copies in the cell and calculated 'relative coverage' as (normalized coverage of each bin) / (reference coverage). (2) To define the ecDNA based on the relative coverage, we investigated the relative coverage of known ecDNA, CNV and random regions in K-562 and MDA-MB-231 cell lines. We selected following ecDNA regions: chr9:130,731,514-131,280,213 and chr22: 23,280,553-23,290,953 in K-562, and chr6: 33,066,224-43,717,063 in MDA-MB-231. CNV regions were selected based on the CCLE dataset. For the 'random region', we randomly selected 500 regions, each of which contains one gene and ±50,000 bp region. The result showed that the relative coverage of ecDNA was apparently higher than CNV and random region (Supplementary Fig. 6b), and we, therefore, discarded the regions with relative coverage ≤6. (3) In order to improve confidence, only high-coverage bins presented in more than 15 cells were retained. Afterward, consecutive bins were merged. (4) Most of the human genome is tightly packed into complex hierarchical structures, while ecDNA shows relatively more accessible chromatin. To test this, we evaluate the accessibility of ecDNA in K562 and MDA-MB-231 cell lines using a TSS enrichment score (Supplementary Fig. 6c), which was calculated via signal-noise-ratio, namely, (fragments count within ±50 bp region centered at TSS position)/ (fragments count within ±2000bp region centered at TSS position). Finally, we used TSS enrichment score <0.25 as another condition to separate the candidate ecDNA amplicon region from CNV and random region. In addition, given the sparsity of scATAC-seq data, even an ecDNA derived from a continuous genomic region could be detected as more than one ecDNA fragment based on scATAC-seq data if the region was not evenly and sufficiently covered by scATAC-seq reads due to experimental and/or mapping biases. Therefore, the number of the ecDNA fragments that we have identified should be higher than the number of different ecDNA species found in the cell. To explore this possibility, we investigate the correlation of the coverage across different ecDNA fragments identified in individual cell lines by constructing the ecDNA fragment × cell matrix, with row as ecDNA fragments, column as single cell, and the value as coverage of the corresponding ecDNA fragment in the corresponding cell. Interestingly, ecDNA fragments pairs located in the vicinity (<1MB) often showed a high correlation (average Pearson correlation coefficient (PCC) equal to 0.4) than those from distant regions (>1MB) (average PCC equal to 0), as exemplified by SF268 cell line (Supplementary Fig. 6e). The finding to a large extent supports our hypothesis. Then we integrated the ecDNA fragments using an unsupervised hierarchical clustering method. First, we calculated the correlations between ecDNA fragments using the cell x ecDNA coverage matrix, where cells are represented as columns, ecDNA as rows, and coverage values as elements. Subsequently, hierarchical clustering was performed based on this correlation matrix. To ensure higher similarity of ecDNA coverage within clusters, we set the number of clusters to one-third of the total ecDNA fragments. If more than 70% of ecDNA fragments in one cluster showed PCC greater than 0.3 and were derived from neighboring chromosome regions, they would be merged.

### Hypoxia data analysis

Limma (v.3.50.0) was used to analyze differential expression between normoxia and hypoxia treatment[57]. The genes with adjusted p-value < 0.01 and log2FC > 2 were selected as significant differentially expressed genes in each cell line for further analysis.

To match cell subclusters before and after hypoxia treatment within the individual cell line, first, Seurat's FindIntegrationAnchors was performed to identify anchors between the two datasets. Then, IntegrateData was used to integrate cells from two conditions.

Clustering was performed on all cells from the integrated data. Then we matched the cell subclusters in two conditions with cell subclusters in the integrated data. If more than 80% of cells in the subclusters in two conditions are clustered together in the integrated data, these two subclusters were matched in two conditions.

To evaluate the response of both the entire cell lines and their subpopulations to hypoxia, we employed the AddModuleScore function of Seurat to score their HALLMARK_HYPOXIA and publicly available HIF signature[58]. To evaluate how different cell subclusters responded differently, differential gene expression analysis and gene set enrichment analysis were applied to the matched subclusters. To explain why different cell subclusters respond differently from the epigenetic perspective, we apply the FindTransferAnchors function in ArchR to integrate scATAC-seq data before hypoxia treatment with scRNA-seq data after hypoxia treatment and identified key TFs by enriching motifs from subgroup-specific accessible peaks.

### Quantification and statistical analysis

All data analyses were conducted in R v.4.2.1 and Python 3.11.4. Statistical analyses are described in the respective "Methods" subsections and are briefly described in the figure legends. P-values were false discovery rate-corrected for multiple hypotheses testing where indicated. $P < 0.05$ was considered statistically significant. All data are presented as mean ± SD unless otherwise indicated.

### Reporting summary

Further information on research design is available in the Nature Portfolio Reporting Summary linked to this article.

### Data availability

The bulk cancer cell line RNA-seq data used in this study are publicly available through depmap portal at https://depmap.org/portal and GEO (SKBR3: under accession code GSE7562 at https://www.ncbi.nlm.nih.gov/geo/query/acc.cgi?acc=GSE7562; LS174T: under accession code GSE18560 at https://www.ncbi.nlm.nih.gov/geo/query/acc.cgi?acc=GSE18560). The raw scRNA-seq and scATAC-seq data generated in this study have been deposited in China National GeneBank DataBase (CNGBdb) Sequence Archive (CNSA) with accession number CNP0004330 and also in the Genome Sequence Archive (GSA) database[59,60] under accession number PRJCA021248. The processed scRNA-seq and scATAC-seq data generated in this study are available in CNSA with accession number CNP0003658 and also in GSA under accession number PRJCA020910. The CCLA website can be accessed at https://db.cngb.org/cdcp/scatlashcl/, which is an open and interactive database for exploration. The remaining data are available within the article, Supplementary Information or Source Data file. Source data are provided with this paper.

### Code availability

Custom code used for this paper is available from GitHub at https://github.com/liushang17/CCLA[61].

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

## Acknowledgements

We are grateful to Xudong Zhou and Xiyang Chen for helpful suggestions on computational analysis; Jichang Wang and Zhongjian Chen for kindly sharing human cancer cell lines; the Center for Computational Science and Engineering of SUSTech for the support on the computational resources. This work was supported by the National Key Research and Development Program of China (to W.C.: 2021YFF1201000), the National Natural Science Foundation of China (to Q.Z.: 32100613), the National Key Research and Development Program of China (to W.C.: 2022YFC3400400), Shenzhen Science and Technology Program (to W.C.: KQTD20180411143432337; to Q.Z.: JCYJ20210324104605014; to L.F.: JCYJ20190809154407564;), Shenzhen Key Laboratory of Gene Regulation and Systems Biology (to W.C.: ZDSYS20200811144002008), SUSTech Research Foundation (to W.C.), Guangdong Basic and Applied Basic Research Foundation (to L.W.: 2021A1515110832), Guangdong-Hong Kong Joint Laboratory on Immunological and Genetic Kidney Diseases (to S.-P.Liu: 019B121205005), Shenzhen Key Laboratory of Single-Cell Omics (to S.-P.Liu: ZDSYS20190902093613831), and China National GeneBank.

## Author contributions

W.C., Q.Z., L.F., L.W. and Y.Hou conceived the study. Q.Z. and L.F. designed the experiments. Q.Z., Y.Li, J.H. and Y.T. prepared the cell lines. Q.Z. and L.F. constructed the plasmid. Y.Z., Y.Li., J.H., Z.S., S.W., D.D., C.W., Y.Huang, S.J., and Y.Y. processed single-cell experiments with help from C.L. X.Zhao, Y.Z., S.Liu, and C.T. designed and implemented computational strategies with Q.Z., L.W., L.F., and analyzed data with help from G.L., X.Zou, Z.L. and M.Z. T.Y., T.L., Y.Liu and W.Y. developed an online database. Q.Z. and L.F. wrote the paper with input from X.Zhao. and Y.Z. L.W., W.C., Y.Hu, H.C., S.W., X.C. and Y.Li. reviewed and revised the manuscript. X.J., A.C. and X.X. discussed the manuscript. Q.Z., L.W., W.C., L.F., L.L. and S.-P.Liu coordinated and supervised the study.

## Competing interests

The authors declare no competing interests.
