## [Peer Review File · Nature Communications]

Single cell multi-omics reveal intra-cell-line heterogeneity across human cancer cell linesReviewers' Comments:

Reviewer #1:

Remarks to the Author:

The rationale for this study from Zhu et al., is outlined as the need to better understand intra-cell line heterogeneity to help identify suitable models for research into specific phenotypes. Additionally, they also aim to address a gap in our knowledge concerning the molecular mechanisms of intra-cell line heterogeneity across lines derived from multiple tumour types.

The authors perform multiple sequencing modalities on around 45 cell lines in this study, selecting cells from across a number of tumour types, with heavy weighting towards colo-rectal and breast cancer origins. While they document intra-cell line variability comprehensively in this manuscript, they have not succeeded in their aim of providing molecular mechanisms for this as while they suggest transcription factor binding that is likely to be occurring differentially, the mechanisms leading to this remain correlative in this study.

The data within this manuscript is likely to represent a useful tool in the field, but while a description of transcriptional heterogeneity in vitro is interesting, the manuscript is descriptive, and doesn't add any mechanistic detail to suggest why and how the heterogeneity arises. For example, why are specific transcription factors showing increased activity in specific subclusters of cell lines? Are the oncogenes observed in eDNAs within subclusters associated with specific TF profiles?

In the hypoxia study, the authors use a method to match subclusters before and after hypoxic treatment. However, given that the subclusters may be temporally unstable, is this a reasonable hypothesis – could it not be that the different subclusters react entirely differently to hypoxia?

It is possible to achieve hypoxic microenvironments within normoxic culture conditions, which is highly dependent on the confluence of cells at the point of harvesting. In highly confluent cell cultures, hypoxic gene expression can be observed. This could be a reason for the lack of gene expression changes in one of the subclusters (DLD-1, 0 to 0-2). It isn't clear from the methods if the authors grew the cells to confluence or sub-confluence before harvest.

It is somewhat worrying that hypoxia inducible factor (HIF) transcription factor signatures were not apparent in the scRNA-seq data from hypoxia. Indeed, the comparison of scATAC-seq performed in normoxia against scRNA-seq data in hypoxia does not appear reasonable given the significant changes in chromatin accessibility observed in hypoxia.

Minor points.

1. Line 167 onwards – while many cell lines diversity score and classification were well correlated, some weren't. Could the authors comment on why this might be the case?
2. More detail is required in the Methods section – for example, after hypoxic treatment it is important to know how the cells were processed (washing, etc.) as time during re-oxygenation may produce artefacts.

Reviewer #2:

Remarks to the Author:

The authors generated single cell RNA-seq and ATAC-seq data from human cancer cell lines to illustrate both transcriptomic and epigenetic heterogeneity within individual cell lines. The Authors performs transcriptomics analysis, copy number variation analysis, regulatory analysis, ecDNA analysis, differential transcriptomics analysis under hypoxia treatment. I have some concerns about the manuscript.

1. Since scRNA data of cancer cell line have been generated and analyzed in previous literatures, I think the novelty of this manuscript should be the chromatin accessibility atlas and the hypoxia treatment data. This novelty has not been highlighted so that the contribution of this paper becomes

not clear.

2. Three cell lines from different lineages were pooled for each run. How these single cells are computationally assigned to the corresponding cell line is not clear. How to make sure this computational assignment is correct, robust, or reasonable.
3. An average of 513 cells per cell line is generated. Considering the sparsity of single cell data, it is not clear if the cell number is sufficient to perform further subpopulation analysis.
4. Many descriptions are not clear and not concrete. For example, "To explore the mechanisms underlying epigenomic heterogeneity, firstly, we sought to infer potential transcription factors (TFs) that drive the differential chromatin accessibility between cell subclusters and focused on common TFs appeared in at least three cell lines (Fig. 5a and Supplementary Table 7)." It is not clear how to infer potential TFs that drive differential chromatin accessibility? It requires a citation to previous literatures or a reference to a specific section in the Methods or supplementary materials.
5. Many statements have a clear logical gap. For example, "The results showed that genes located in CNV regions didn't show obvious heterogeneity in chromatin accessibility among different subclusters, suggesting that heterogeneous chromatin accessibility was not induced by CNV, but was mainly contributed by the epigenetic mechanisms (Supplementary Fig. 5 a-b)." "Genes in CNV regions didn't show heterogeneity in chromatin accessibility" and "chromatin accessibility was not induced by CNV" are totally different/independent two events, and there is no clear logical relation between them.

Reviewer #3:

Remarks to the Author:

The authors have done a great deal of work integrating scRNA seq, scATAC-seq, CNV analyses, analyses of ecDNA, and chromatin tracing of human cancer cell lines. They suggest that these results highlight the intratumoral heterogeneity of cancer and define some of its key features.

Overall the analyses are interesting. A number of concerns are raised, which should be considered to strengthen the paper.

What are the key conclusions that advance the field? It is now well-recognized the intratumoral heterogeneity is a major issue, and also that ecDNA contributes. These are important conclusions. However, the paper would be strengthened by providing more specific conclusions. What are the key take away points of this study?

scRNA-seq

1. The analysis is quite solid partly because the methodology is well established.
2. The method for measuring diversity is weak, given by the radius of a low dimensional projection of the cells. Why did the authors use PCA, when they had already done a UMAP. You can have large clusters without their being separate.
3. The NMF used to get the main gene expression programs looks good, and the clusters make intuitive sense.

scATAC-seq

1. The classification seems somewhat Indiscriminate and is not properly explained.

CNV analysis

1. They choose to do CNV analysis using RNA, but the method is not well-supported. They need to validate their method on known CNVs in these cell-lines by comparing with the bulk genomic data. The sum of sc CNV data inferred by them should ideally match the bulk DNA CNV data.
2. Chromatin accessibility work is good, but some of the conclusions seem to be overstated. For example, they discuss that FOXA2 target genes are involved in KRAS signaling, but none of these associations rise to the accepted level of significance (Figure 5i)

ecDNA

1. Their ecDNA calls raise some issues because a median of 270 ecDNA per cell-line is way above what others have reported, and there is no validation. It is possible that the high number is due to seeing the same region repeatedly in multiple single-cells in one cell-line? If so, they need to explain how many distinct such regions are found per cell-line. Many regions might contribute to one ecDNA and they do not consider that. They do not show any kind of specificity or sensitivity of detection. Perhaps they can call these 'high-accessibility regions' and mention that they sometimes overlap with known ecDNA.

chromatin tracing

1. When they propagate cells, the scRNA seq clusterings change. That is not very surprising and could be caused by many factors including sampling variation. What controls are used?

Point-by-point response to the referees' comments

The general response to the reviewers:

We thank all the referees for their time and appreciate their highly valuable comments and suggestions, which helped us to improve our manuscript. During the revision, we have carefully addressed all the points raised by them, as shown below marked in red.

Reviewer #1 (Remarks to the Author):

The rationale for this study from Zhu et al., is outlined as the need to better understand intra-cell line heterogeneity to help identify suitable models for research into specific phenotypes. Additionally, they also aim to address a gap in our knowledge concerning the molecular mechanisms of intra-cell line heterogeneity across lines derived from multiple tumour types.

The authors perform multiple sequencing modalities on around 45 cell lines in this study, selecting cells from across a number of tumour types, with heavy weighting towards colorectal and breast cancer origins. While they document intra-cell line variability comprehensively in this manuscript, they have not succeeded in their aim of providing molecular mechanisms for this as while they suggest transcription factor binding that is likely to be occurring differentially, the mechanisms leading to this remain correlative in this study.

Answer:

We thank the referee for the comment. The highlight of our study includes: 1) As a resource, it offers a comprehensive dataset for the analysis of intra-cell-line heterogeneity at both the transcriptomic and epigenomic levels. Whereas in a previous study, the scRNA-seq has been performed for 198 cancer cell lines¹, the scATAC-seq for cancer cell lines at this scale is for the first time. 2) We unveil the diverse and dynamic nature of cellular states within individual cell lines, shedding light on the intricacies of cell-to-cell variability. On one hand, we identify genetic variations, including copy number variations (CNVs) and extrachromosomal circular DNAs (ecDNAs), as contributing factors to transcriptomic heterogeneity. Here, in particular, we developed a new method to detect ecDNA based on scATAC-seq data. On the other hand, we demonstrated the impact of transcription factor

(TF) activity on epigenetic heterogeneity. Our findings highlight the interplay between genetic and epigenetic factors in shaping cellular heterogeneity. 3) Through rigorous hypoxia treatment experiments, we demonstrate the heterogeneous cellular response to hypoxia from the same cell line, even from the same subpopulation. The latter led us to the identification of differential primed chromatin states driven by specific TFs preceding hypoxia. 4) Our genetic barcoding combined with scRNA-seq allows us to track individual cell trajectories. Our data demonstrated the dynamic nature of cellular states. The observation that cells can transition between different transcriptional states over time, demonstrated the important contribution of epigenetic plasticity in shaping cellular heterogeneity. In summary, our study advances the field by providing a comprehensive understanding of the factors influencing intra-cell-line heterogeneity and highlighting the dynamic and complex nature of cellular states within tumors. These findings contribute valuable insights into tumor biology and have implications for the further development of precision medicine and targeted therapies.

However, while our study provides valuable insights into the molecular features associated with intratumoral heterogeneity in cancer cell lines, we acknowledge the limitations of our study in establishing the detailed underlying molecular mechanisms. We thank the referee for raising this point, and we will take his/her feedback into consideration when planning future investigations to further elucidate the detailed mechanisms driving transcriptomic heterogeneity in cancer cell lines.

The data within this manuscript is likely to represent a useful tool in the field, but while a description of transcriptional heterogeneity in vitro is interesting, the manuscript is descriptive, and doesn't add any mechanistic detail to suggest why and how the heterogeneity arises. For example, why are specific transcription factors showing increased activity in specific subclusters of cell lines? Are the oncogenes observed in ecDNAs within subclusters associated with specific TF profiles?

Answer:

We appreciate the valuable suggestions provided by the referee. Our previous analysis has revealed that the activity of TFs is influenced by various factors, including their expression levels and their promoter accessibility (Fig. 5). Here we further explored how the activity of specific TFs is affected by their presence within specific genomic structures such as ecDNA. Across all the cell lines studied,

a diverse range of TFs (4 to 24) were found to be associated with ecDNA (Fig. R1a, also as Supplementary Fig. 6i in the revised version). Here, to demonstrate that the presence of ecDNA could have an impact on the expression of its encompassed genes, including the transcription factors, which in turn are associated with cellular heterogeneity, we took SNB75 cell line as an example. As shown in Fig. R1b, by utilizing transcriptome and chromatin accessibility data, we identified five distinct sub-clusters (R0-R4) within SNB75 (Fig. R1b, also as Figure 5b and c). Notably, the ecDNA region (chr22:37439388_38039388) exhibited a significantly higher copy number in sub-cluster R3 compared to the other clusters (Fig. R1c, also as Supplementary Fig. 6j in the revised version). Gene set variation analysis (GSVA), which could be used to estimate the pathway activity in each cell by transforming a gene-by-sample expression data matrix into a corresponding gene-set-by-sample expression data matrix, revealed that all the 29 genes within the ecDNA region (chr22:37439388_38039388) exhibited significantly higher expression levels in sub-cluster R3 (Fig. R1d, also as Supplementary Fig. 6k in the revised version). Moreover, we observed that *SOX10*, a TF located within the ecDNA region (chr22:37439388_38039388), displayed a higher expression level specifically in sub-cluster R3 (Fig. R1e, also as Supplementary Fig. 6l in the revised version). Importantly, its target genes were also found to be significantly more activated in sub-cluster R3 (Fig. R1f, also as Supplementary Fig. 6m in the revised version). These results demonstrated that the presence of ecDNA could impact the activity of specific transcription factors, ultimately contributing to cellular heterogeneity.

Figure R1. (a) The number distribution of TFs within ecDNAs for subclusters across 39 cell lines. (b) UMAP plot scATAC-seq (A1-A5) and scRNA-seq (R0-R4) clusters of SNB75. (c) The copy number of ecDNA (chr22:37439388_38039388) across the five scRNA-seq subclusters of SNB75. (d) The GSVA score of gene sets located on ecDNA (chr22:37439388_38039388) in different scRNA-seq subclusters of SNB75. (e) The expression level of *SOX10* in different scRNA-seq subclusters of SNB75. (f) The TF activity of *SOX10* in different scRNA-seq subclusters of SNB75. The TF activity is represented by the area under the recovery curve (AUC) score calculated by AUCcell in SCENIC workflow². In essence, the AUC score reflects the enrichment of the TF and its putative targets in a particular cell.

As suggested by the referee, we further conducted an analysis to identify transcription factors (TFs) that target the oncogenes located within extrachromosomal circular DNA (ecDNA) across the investigated cell lines. Here, the oncogenes, which were defined based on COSMIC database, located within ecDNA in the sub-clusters were extracted. Metascape³ was further used to predict the TFs targeting these oncogenes. A total of 78 TFs were found to target the oncogenes within ecDNAs across the 15 cell lines (Fig. R2a). These TFs were sorted based on the number of cell lines in which they regulate the ecDNA oncogenes, and the top 5 TFs were TP53, ING4, ABL1, PML, and BRCA1 (Fig. R2a). Notably, these TFs are known to be active in cancer cell lines and widely

expressed, such as TP53 and BRCA1 (Fig. R2b) ⁴. Then, we further analyzed the impact of these TFs targeting the ecDNA oncogenes, using the MDA-MB-231 cell line as an example. The oncogenes located on ecDNAs within sub-cluster A1 were found to be significantly associated with TP53 (Fig. R2c, d). Consistent with this, the expression as well as the activity of TP53 also exhibited significantly higher levels in R0 compared to R1 and R2 (Fig. R2e, f). In summary, these results indicate that certain TFs could influence the expression of oncogenes within ecDNAs of sub-clusters, thereby contributing to cellular heterogeneity. These findings shed light on the potential regulatory mechanisms underlying the dynamics of ecDNA-associated oncogenes and their impact on cellular heterogeneity.

Figure R2. (a) TFs sorted with the number of cell lines in which they regulate ecDNA oncogenes. Each dot refers to TFs targeting ecDNA oncogenes across cell lines. (b) Expression level of the top five TFs in different cell lines. (c) UMAP plot scATAC-seq (A1-A4) and scRNA-seq (R0-R2) clusters of MDA-MB-231. (d) Bar plot shows the significant enrichment of TP53 as the upstream TFs regulating the ecDNA oncogenes of A1 (R0) cluster in MDA-MB231. (e) The expression

distribution of *TP53* in different scRNA-seq clusters of MDA-MB-231. P value was calculated by the Student's T test. (f) The GSVA score of gene sets targeted by the TF TP53 in different scRNA-seq clusters of MDA-MB-231. P value was calculated by the Student's T test.

In the hypoxia study, the authors use a method to match subclusters before and after hypoxic treatment. However, given that the subclusters may be temporally unstable, is this a reasonable hypothesis – could it not be that the different subclusters react entirely differently to hypoxia? It is possible to achieve hypoxic microenvironments within normoxic culture conditions, which is highly dependent on the confluence of cells at the point of harvesting. In highly confluent cell cultures, hypoxic gene expression can be observed. This could be a reason for the lack of gene expression changes in one of the subclusters (DLD-1, 0 to 0-2). It isn't clear from the methods if the authors grew the cells to confluence or sub-confluence before harvest.

Answer:

As suggested by the referee, we have now added more details of the method, in which we emphasized that cells were collected when they were 80-90% confluent (See methods in the revised version). We also added analysis on hypoxia inducible factor (HIF) transcription factor signatures between before and after hypoxic treatment based on scRNA-seq data⁵, which showed that hypoxia related genes are significantly changed after hypoxic treatment (Fig. R3a, also as Supplementary Fig. 8a in the revised version). Notably, despite the influence of hypoxia, the distinct properties and specific features of different sub-clusters were maintained across both normal and hypoxia conditions, as evidenced by the significant overlap between the differentially expressed genes (DEGs) identified between N_subclusters, and H_subclusters (Fig. R3b, also as Supplementary Fig. 8b). Furthermore, under hypoxia conditions, all subclusters of DLD-1 exhibited higher expression levels of hypoxia signature genes compared to the subclusters under normal conditions (Fig. R3c, also as Supplementary Fig. 8c in the revised version). Within the hypoxia condition, sub-cluster H_1 demonstrated a higher expression of HIF signature genes compared to H_0, suggesting varying levels of response among different clusters with distinct features (Fig. R3c, also as Supplementary Fig. 8c in the revised version). Moreover, sub-cluster H_0-1 showed higher expression of HIF signature and hypoxia score compared to H_0-2, suggesting cells within one cluster may respond

differently to the hypoxia (Fig. R3c, also as Supplementary Fig. 8c in the revised version). These findings together indicate that on one hand, the sub-clusters identified under normal conditions could maintain some of their distinct profile even under hypoxic conditions. On the other hand, cells with distinct features could differentially respond to hypoxia.

Figure R3. (a) Violin plot showing the GSVA score of HIF signature between normal and hypoxia condition across ZR-75-1, DLD-1, and SW620. A two-sided Wilcoxon test was used to assess statistical significance. (b) The overlap of DEGs identified between different sub-clusters under normoxia and DEGs identified between different sub-clusters under hypoxia in ZR751, DLD1, and SW620. A hypergeometric test was used to test the statistical significance. (c) The GSVA score distribution of Hypoxia and HIF signature across the sub-cluster under Normal and Hypoxia conditions in DLD1. A two-sided Wilcoxon test was used to assess statistical significance.

It is somewhat worrying that hypoxia inducible factor (HIF) transcription factor signatures were not apparent in the scRNA-seq data from hypoxia. Indeed, the comparison of scATAC-seq performed

in normoxia against scRNA-seq data in hypoxia does not appear reasonable given the significant changes in chromatin accessibility observed in hypoxia.

Answer:

As suggested by the referee, we have now added analysis on HIF transcription factor signatures between before and after hypoxic treatment based on scRNA-seq data, which showed that HIF signatures are significantly changed after hypoxic treatment (Fig. R3a, also as Supplementary Fig. 8a in the revised version).

Our comparison of scATAC-seq performed in normoxia against scRNA-seq data in hypoxia only focus on the cell lines whose cells in one sub-cluster responded differently to the hypoxia treatment. Our purpose is to reveal whether primed chromatin states regulated by specific TFs could precede the change of cell state under hypoxia treatment. For this purpose, we compared the scATAC-seq data under normoxia with scRNA-seq data under hypoxia and found some potential TFs that are responsible for the heterogeneous response under hypoxia.

Minor points.

1. Line 167 onwards – while many cell lines diversity score and classification were well correlated, some weren't. Could the authors comment on why this might be the case?

Answer:

As suggested by the referee, we now explained why some cell lines' diversity scores are not completely correlated with classification. The classification and diversity score were computed based on different metrics. Based on the 30 PCs used for profiling the cell atlas, the diversity score⁶ quantifies the distance of each cell from the center of the cell line, i.e. the collective distances within the cell line. In certain cases, the Euclidean distances within clusters are significantly smaller than the distances between clusters. As a result, the cell line showed discrete pattern according to the classification, even though its diversity score could be low if the collective distance is not that large. Vice versa, the collective distance can be large whereas the distance within clusters are not significantly different from the distance between clusters. To illustrate this, we provide examples of four cell lines, each representing one example for the four different scenarios: discrete with a high

diversity score (SNB75), continuous with a high diversity score (LS174T), discrete with a low diversity score (HeLa), and continuous with a low diversity score (Hap1). We calculated the distance between every two cells in these cell lines and plot the distribution, which indicated potential heterogeneity within the cell lines. The presence of double peaks in the density profile suggests the existence of potential discrete type of subpopulations, while the absence of double peaks implies insignificant differences between cell subpopulations. The discrete cell lines (SNB75 and HeLa) display obvious double peaks, whereas the continuous cell lines (LS174T and Hap1) exhibit a single peak (Fig. R4a, b). These results suggest that the diversity and classification provide measurement of cellular heterogeneity from different angles, the insight of which could complement each other.

Figure R4. (a) Diversity scores for different cell lines. Cell lines with light blue indicating continuous pattern and cell lines with dark blue indicating discrete pattern. (b) The density distribution plot of the distance between two cells across four cell lines, representing 4 different situations, including discrete with high diversity score (SNB75), continuous with high diversity score (LS174T), discrete with low diversity score (HeLa), and continuous with low diversity score (Hap1).

2. More detail is required in the Methods section – for example, after hypoxic treatment it is important to know how the cells were processed (washing, etc.) as time during re-oxygenation may produce artefacts.

Answer:

As suggested by the referee, we have now added more details for several experiments and analysis

in the Methods section, including Hypoxia treatment, Cell line assignment, Transcription factor motif enrichment analysis, GSVA analysis, ecDNA detection, and Hypoxia data analysis. As an example, we rewrite Hypoxia treatment in the methods section as follows: The normoxic set of plates was placed in an aerobic incubator. To create hypoxia, the hypoxic set was moved to a Hypoxia Incubator Chamber (StemCell) which was equilibrated at 37°C in a humidified atmosphere of 5% CO₂ and 1% O₂. After 24 hours of incubation, the cells, which were 80%-90% confluent, were washed once with PBS, then harvested with 0.05% Trypsin EDTA. After neutralization with complete medium, centrifugation, and resuspension in PBS with 0.04% BSA, the cells were measured.

Reviewer #2 (Remarks to the Author):

The authors generated single cell RNA-seq and ATAC-seq data from human cancer cell lines to illustrate both transcriptomic and epigenetic heterogeneity within individual cell lines. The Authors performs transcriptomics analysis, copy number variation analysis, regulatory analysis, ecDNA analysis, differential transcriptomics analysis under hypoxia treatment. I have some concerns about the manuscript.

1. Since scRNA data of cancer cell line have been generated and analyzed in previous literatures, I think the novelty of this manuscript should be the chromatin accessibility atlas and the hypoxia treatment data. This novelty has not been highlighted so that the contribution of this paper becomes not clear.

Answer:

As suggested by the referee, we have strengthened the analysis of chromatin accessibility atlas and the hypoxia treatment data in our revised manuscript. Firstly, we performed more analysis on scATAC-seq data and further described the classification and diversity score based on scATAC-seq data (Line 261-269 in the revised manuscript.). Meanwhile, we demonstrated the use of the scATAC-seq data as a resource, which helps to reveal the effect of epigenetic diversity, the identification of ecDNA, and prediction of primed chromatin state before hypoxia treatment and etc. In addition, to rule out the potential technical artifact, as suggested by referee 1, we demonstrated the efficient induction of hypoxia response by comparing the cellular transcriptome between before

and after hypoxic treatment (see Response to Referee 1, point 3, Fig. R3).

2. Three cell lines from different lineages were pooled for each run. How these single cells are computationally assigned to the corresponding cell line is not clear. How to make sure this computational assignment is correct, robust, or reasonable.

Answer:

We apologize for being unclear in this part of the analysis. Tumor cells from different cell lines can be effectively separated by clustering based on scRNA-seq data, given that cells from different cell lines have significant differences in their transcriptome profile (Supplementary Fig. 1b). In addition, the pseudobulk RNA-seq data derived from scRNA-seq data correlated well with the published bulk RNA-seq dataset for the corresponding cell lines (Supplementary Fig. 1f). To further demonstrate the efficiency of our clustering analysis on the cell type identification, we integrated the published bulk RNA-seq dataset⁷ with our scRNA-seq dataset. Taking the batch of CL200136256_L01 as an example, which is the mixture of three cell lines, including HCT116, MCF7, and A253, we merged the published bulk RNA-seq dataset of these three cell lines with our scRNA-seq data. As shown in Fig. R5a, b, not only the three cell lines were separated (Fig. R5a, also as Supplementary Fig. 1c in the revised version), but also the pseudo-single cells deriving from bulk RNA-seq datasets match the corresponding single cell populations in UMAP (Fig. R5b, also as Supplementary Fig. 1d in the revised version). Moreover, the marker genes are significantly expressed only in the given cell lines based on both the bulk and single-cell datasets (Fig. R5c, also as Supplementary Fig. 1e in the revised version). The correlation between the pseudobulk RNA-seq derived from the scRNA-seq and the published bulk RNA-seq data is much higher for the matching cell lines (Fig. R5d, and Supplementary Fig. 1i). These results, together demonstrated that the effectiveness and robustness of our cell line assignment. We added the description for cell line assignment to the Methods section of the revised manuscript.

Figure R5. (a) UMAP plot showing the three cell lines in the batch of CL200136256_L01, labeled by different colors. (b) UMAP plot showing the resource of each spot, with cell in blue from scRNA-seq and red from bulk RNA-seq. (c) Heatmap depicting the expression of the 10 marker genes of the three cell lines based both on scRNA-seq and bulk RNA-seq dataset. (d) Heatmap representing the correlation between pseudobulk RNA-seq derived from scRNA-seq (Y-axis) and published bulk RNA-seq dataset (X-axis).

3. An average of 513 cells per cell line is generated. Considering the sparsity of single cell data, it is not clear if the cell number is sufficient to perform further subpopulation analysis.

Answer:

We thank you for the comment. To address this concern, we performed down-sampling analysis on one cancer cell line dataset with cell number equal to 7693 (Hepa 1.6, unpublished). As shown in Fig. R6a, three subpopulations (C0-2) were identified by clustering analysis with 7693 cells from Hepa1.6 cell line (Fig. R6a). Then, we randomly sampled 500 and 300 cells from these 7693 cells and performed clustering analysis (Fig. R6b, c). The three subpopulations were again clearly separated in the new UMAP based on the UMAP of 500 cells, even of 300 cells (Fig. R6b, c). Furthermore, the subpopulations determined by clustering analysis based on the 500 cells and 300 cells match to the original subpopulations C0-C2 (Fig. R6b, c). Meanwhile, the characteristic gene expression of C0-C2 subpopulations could be well maintained in the cell subpopulations clustered based on the down-sampled 500-cell or 300-cell datasets (Fig. R6d-f). Thus, we believe that a scRNA-seq dataset of 500 cells, even down to 300 cells is effective in identifying cell subpopulations from one cell line.

Figure R6. (a) UMAP plot showing the sub-clusters of Hepa1.6 with cell number equal to 7693. Each sub-cluster was marked with different colors. (b) UMAP plot showing the sub-clusters of Hepa1.6 with cell number down-sample to 500. Cells are marked with the sub-clusters determined by the clustering results with cell number equal to 7693 (Left) and with cell number equal to 500 (Right) in different colors. (c) UMAP plot showing the sub-clusters of Hepa1.6 with cell number down-sample to 300. Cells are marked with the sub-clusters determined by the clustering results with cell number equal to 7693 (Left) and with cell number equal to 300 (Right) in different colors. (d-f) Heatmap showing the expression of differentially expressed genes for the sub-clusters based on 7693 (d), 500 (e) and 300 (f) cells.

4. Many descriptions are not clear and not concrete. For example, “To explore the mechanisms underlying epigenomic heterogeneity, firstly, we sought to infer potential transcription factors (TFs) that drive the differential chromatin accessibility between cell subclusters and focused on common TFs appeared in at least three cell lines (Fig. 5a and Supplementary Table 7).” It is not clear how to infer potential TFs that drive differential chromatin accessibility? It requires a citation to previous literatures or a reference to a specific section in the Methods or supplementary materials.

Answer:

As suggested by the referee, we have now cited the previous literatures and added more details of ‘Transcription factor motif enrichment analysis’ in the Methods section: To identify key TFs responsible for subgroup-specific accessible chromatin regions, two approaches, including ArchR⁸ and ChromVAR⁹, were used for identifying TFs with motifs enriched in differential ATAC peaks from the subgroups of a same cell line. ArchR is employed to infer potential TFs that drive differential chromatin accessibility across cell sub-clusters. More specifically, for ArchR, addMotifAnnotations was used to determine the motif presented in the peak set and peakAnnoEnrichment was used to identify the motifs enriched in differentially accessible peaks. Based on the motif determined by addMotifAnnotations, ChromVAR was applied to predict the enrichment of TF activity on a per-cell basis. Initially, the chromVAR::getBackgroundPeaks function was used to get background peaks based on similarity in GC-content and the number of fragments across all samples using the Mahalanobis distance, then the addDeviationsMatrix function was used to compute per-cell deviations across all of our annotated motifs. The literature cited are ‘ArchR is a scalable software package for integrative single-cell chromatin accessibility analysis’ and ‘chromVAR: inferring transcription-factor-associated accessibility from single-cell epigenomic data’.

5.Many statements have a clear logical gap. For example, “The results showed that genes located in CNV regions didn’t show obvious heterogeneity in chromatin accessibility among different subclusters, suggesting that heterogeneous chromatin accessibility was not induced by CNV, but was mainly contributed by the epigenetic mechanisms (Supplementary Fig. 5 a-b).” “Genes in CNV regions didn’t show heterogeneity in chromatin accessibility” and “chromatin accessibility was not induced by CNV” are totally different/independent two events, and there is no clear logical relation between them.

Answer:

We apologize for the confusion. In the revised manuscript, we rewrite this part as following: Detection of increased open chromatin could be attributed to either increased chromatin accessibility or increased DNA copy number. To exclude the influence caused by the latter, i.e. CNVs, we demonstrated that the chr15 and chr9, which showed CNV in sub-clusters of SNB75 and

MDA-MB231 cell line respectively, didn't show evident heterogeneity in chromatin accessibility among different sub-clusters (Supplementary Fig. 5a-b). This suggested that heterogeneous chromatin accessibility and CNV could independently contribute to transcriptomic heterogeneity, and for these three cell lines, chromatin accessibility contributes considerably to transcriptomic heterogeneity.

Reviewer #3 (Remarks to the Author):

The authors have done a great deal of work integrating scRNA seq, scATAC-seq, CNV analyses, analyses of ecDNA, and chromatin tracing of human cancer cell lines. They suggest that these results highlight the intratumoral heterogeneity of cancer and define some of its key features.

Overall the analyses are interesting. A number of concerns are raised, which should be considered to strengthen the paper.

What are the key conclusions that advance the field? It is now well-recognized that intratumoral heterogeneity is a major issue, and also that ecDNA contributes. These are important conclusions. However, the paper would be strengthened by providing more specific conclusions. What are the key take away points of this study?

Answer:

We appreciate the referee's recognition of the contribution of our work to the research of intratumoral heterogeneity. The highlight of our study includes: 1) as a resource, it offers a comprehensive dataset for the analysis of intra-cell-line heterogeneity at both the transcriptomic and epigenomic levels. Whereas in a previous study, the scRNA-seq has been performed for 198 cancer cell lines¹, the scATAC-seq for cancer cell line at this scale is for the first time. 2) We unveil the diverse and dynamic nature of cellular states within individual cell lines, shedding light on the intricacies of cell-to-cell variability. On one hand, we identify genetic variations, including copy number variations (CNVs) and extrachromosomal circular DNAs (ecDNAs), as contributing factors to transcriptomic heterogeneity. Here, in particular, we developed a new method to detect ecDNA based on scATAC-seq data. On the other hand, we demonstrated the impact of transcription factor (TF)

activity on epigenetic heterogeneity. Our findings highlight the interplay between genetic and epigenetic factors in shaping cellular heterogeneity. 3) Through rigorous hypoxia treatment experiments, we demonstrate that the heterogeneous cellular response to hypoxia from a same cell line, even from a same subpopulation. The latter led us to the identification of differential primed chromatin states driven by specific TFs preceding hypoxia. 4) Our genetic barcoding combined with scRNA-seq allows us to track individual cell trajectories. Our data demonstrated the dynamic nature of cellular states. The observation that cells can transit between different transcriptional states over time, demonstrated the important contribution of epigenetic plasticity in shaping cellular heterogeneity. In summary, our study advances the field by providing a comprehensive understanding of the factors influencing intra-cell-line heterogeneity and highlighting the dynamic and complex nature of cellular states within tumors. These findings contribute valuable insights into tumor biology and have implications for the further development of precision medicine and targeted therapies.

scRNA-seq

1. The analysis is quite solid partly because the methodology is well established.
2. The method for measuring diversity is weak, given by the radius of a low dimensional projection of the cells. Why did the authors use PCA, when they had already done a UMAP. You can have large clusters without their being separate.
3. The NMF used to get the main gene expression programs looks good, and the clusters make intuitive sense.

Answer:

As for the point 2, both PCA and UMAP are unsupervised dimensionality reduction algorithms that can be used to discover structures in high-dimensional data. UMAP is a non-linear method that aims to preserve the local structure between data points in a low-dimensional space while ignoring the global structure. Given its preservation of local structure, UMAP is now one of the most commonly used tool for data visualization. In comparison, PCA is a linear method, which performs orthogonal transformation on the data in the original feature space to obtain a new low-dimensional feature space, and retains more global information about the original data. Diversity score was defined in a

previous study¹⁰, in which Ma et al. applied PCA for dimension reduction and then in order to capture major information and reduce noise, used top 30 PCs instead of original gene expression profiles of the cells to measure diversity of each sample. Given that the reliability and robustness of this metric have been well established¹⁰, we directly employed their strategy.

scATAC-seq

1. The classification seems somewhat Indiscriminate and is not properly explained.

Answer:

As suggested by the referee, we further explained the challenge of clustering analysis based on scATAC-seq and added more analysis. First, in contrast to scRNA-seq data, the scATAC-seq data is highly sparse and therefore it was challenging, to define distinct cellular clusters based on scATAC-seq data using UMAP, even for those cell lines with distinct subclusters revealed based on scRNA-seq data. To make a compromise, we categorized the cell lines into two groups based on the shape of the cell populations projected in UMAP: 'indiscriminate' and 'differential' cell lines. For 24 'indiscriminate' cell lines (e.g., COLO 205), cells exhibited no evident clusters in chromatin accessibility (Fig. R7a, also as Fig. 3c in the revised version). In contrast, a total of 15 'differential' cell lines (e.g., MDA-MB-231 and SF268) displayed distinct clusters, indicating considerably higher inter- than intra-cluster variability in chromatin accessibility (Fig. R7a, also as Fig. 3c in the revised version).

In the revised manuscript, we added the application of diversity score to systematically quantify the intra-cell-line heterogeneity of each cell line based on scATAC-seq data. Similarly, as done for scRNA-seq data, we employed the principal component value from ArchR analysis result to project all cells captured in scATAC-seq to an eigenvector space and defined the centroids of the individual cell line, and then calculated the diversity score as the average distance to their specific centroids within individual cell lines. In general, the indiscriminate/differential classification correlated well with the diversity score, and the diversity score of cell lines belonging to the differential group was significantly higher than those from indiscriminate groups (Fig. R7b and c, also as Fig. 3d and e in the revised version), suggesting cell lines with a high degree of heterogeneity often derives distinct subclusters. A similar trend was also observed based on scRNA-seq data.

Figure R7. (a) UMAP plot illustration of two types of cell lines: with (differential, MDA-MB-231 and SF268) or without (indiscriminate, COLO 205) distinct clusters of chromatin accessibility. (b) Diversity scores for different cell lines. Cell lines with light blue indicate indiscriminate pattern and cell lines with red indicate differential pattern. (c) A violin plot depicting the diversity score of the cell lines with the two patterns (differential vs indiscriminate), and a one-sided Wilcoxon test was used to test the statistical significance (p value=0.045).

CNV analysis

1. They choose to do CNV analysis using RNA, but the method is not well-supported. They need to validate their method on known CNVs in these cell-lines by comparing with the bulk genomic data. The sum of sc CNV data inferred by them should ideally match the bulk DNA CNV data.

Answer:

We agree to the referee that CNV analysis based on RNA-seq is not as robust as that based on DNA-seq. Nevertheless, InferCNV was an established tool for CNV analysis based on RNA-seq data and has widely been used to determine the CNV profiles of tumor subpopulations and intra-tumor CNV

heterogeneity. Developed in a primary glioblastoma study, the authors demonstrated that CNV calling by RNA-seq has high correlation ($r=0.72$) with CNV calling by SNP array on 1046 CCLE samples, indicating the accuracy of this method¹¹. More recently, additional studies further validated that the CNV inferred from scRNA-seq using InferCNV matched well with the CNV based on DNA-seq data¹²⁻¹⁶.

As suggested by the referee, we also compared the CNV inferred from scRNA-seq with the CNV estimated based on DNA-seq data, which is available in CCLE database or NCI60 database (data of 35 cell line available). Correlation analysis was applied at arm-level of each chromosome, as done before¹⁴. As shown in Fig. R8a, the Pearson correlation coefficient was higher than 0.5 for the vast majority of cell lines (86%) and higher than 0.7 in half of the cell lines (Fig. R8a). Taking Hs 578T as an example, the gain of chromosome 1p, 6p and 8q, and loss of chromosome 12p could be identified by both methods (Fig.R8 b-c). Taken together, these results supported that the CNVs inferred based on scRNA-seq data significantly match those based on bulk DNA-seq data.

Figure R8. (a) The distribution of Pearson correlation between the CNVs from bulk DNA data and the CNVs from scRNA-seq among 35 cell lines that was available in CCLE database or NCI60 database. (b) The Pearson correlation of mean copy number (CN) ratio between scRNA-seq and bulk DNA data of Hs 578T. (c) Large-scale CNVs inferred from scRNA-seq (top) and identified using WES (bottom) respectively, for a representative cell line Hs 578T.

2. Chromatin accessibility work is good, but some of the conclusions seem to be overstated. For example, they discuss that FOXA2 target genes are involved in KRAS signaling, but none of these

associations rise to the accepted level of significance (Figure 5i)

Answer:

We sincerely appreciate the referee for raising the question regarding statistical significance, particularly in reference to Figure 5h. The target genes of FOXA2 were used to perform functional enrichment analysis by the clusterProfiler package, utilizing the Hallmark gene set. Initially, six terms were obtained with a p-value cutoff set to 0.05 (p value is 0.0068 for KRAS_signaling_up, Fig. R9). Then we used q-value for multiple testing correction to control the false discovery rate. As shown in Figure 5h, the downstream targets of FOXA2 showed higher correlation with KRAS signaling compared to other terms. But as the referee mentioned, the association didn't rise to the level of statistical significant (q value is 0.074, Fig. 5h). We now modified our description based on this result.

Figure R9. Functional enrichment analysis of *FOXA2* downstream target genes with Hallmark gene set. A hypergeometric test was used to test the statistically significant differences. P value < 0.05.

ecDNA

1. Their ecDNA calls raise some issues because a median of 270 ecDNA per cell-line is way above what others have reported, and there is no validation. It is possible that the high number is due to seeing the same region repeatedly in multiple single-cells in one cell-line? If so, they need to explain how many distinct such regions are found per cell-line. Many regions might contribute to one ecDNA and they do not consider that. They do not show any kind of specificity or sensitivity of detection. Perhaps they can call these 'high-accessibility regions' and mention that they sometimes

overlap with known ecDNA.

Answer:

We thank the referee for pointing out the concern about the accuracy of our scATAC-based ecDNA calling. Our current strategy for ecDNA detection primarily focuses on identifying ecDNA fragments from a continuous region of a same chromosome, whereas in the cell one ecDNA could consist of DNA fragments from non-continuous regions of a same chromosome or even from different chromosomes. In addition, given the sparsity of scATAC-seq data, even an ecDNA derived from a continuous genomic region could be detected as more than one ecDNA fragments based on scATAC-seq data if the region was not evenly and sufficiently covered by scATAC-seq reads due to experimental and/or mapping biases. Therefore, the number of the ecDNA fragments that we have identified should be higher than the number of different ecDNA species found in the cell. To explore this possibility, we investigate the correlation of the coverage across different ecDNA fragments identified in individual cell line by constructing the ecDNA fragment \times cell matrix, with row as ecDNA fragments, column as single cell, and the value as coverage of the corresponding ecDNA fragment in corresponding cell. Interestingly, ecDNA fragment pairs located in vicinity ($< 1\text{MB}$) often showed a high correlation (average Pearson correlation coefficient (PCC) equal to 0.4) than those from distant regions ($> 1\text{MB}$) (average PCC equal to 0), as exemplified by SF268 cell line (Fig. R10a, also as Supplementary Fig. 6e in the revised version). The finding to a large extent supports our hypothesis. To merge the neighboring ecDNA fragments showing high correlation of their read coverage, we applied an unsupervised hierarchical clustering method, which was described in detail in the ecDNA detection of the methods section. Briefly, correlations between ecDNA fragments were calculated using the cell*ecDNA coverage matrix. Hierarchical clustering was applied to this correlation matrix. To ensure higher similarity of ecDNA coverage within clusters, we set the number of clusters to one-third of the total ecDNA fragments. If more than 70% of ecDNA fragments in one cluster showed PCC greater than 0.3 and were derived from neighboring chromosome regions, they would be merged. Consequently, the median number of potential ecDNAs across all cell lines decreased from 278 to 229 (Fig. R10b). The new number could still be overestimated because only the ecDNAs from neighboring regions were merged.

To further confirm the accuracy of ecDNA-detection, we applied Circle_finder, a tool to detect

ecDNA based on bulk ATAC-seq. This tool identifies ecDNA based on the detection of junction sites of ecDNA¹⁷. For this purpose, we created pseudobulk ATAC-seq data, one for each cell line. As a result, reads containing junction sites were detected in 13 ecDNA amplicon regions of 5 cell lines (Fig. R10e). However, due to the short read length (50 bp) and high data sparsity of scATAC-seq, the probability of detecting reads containing junction sites is relatively small. Therefore, we did not use junction site reads for further filtering as in bulk-sequencing-based approaches.

FISH experiments are currently the gold standard for validating ecDNA. In the previous study^{18,19}, researchers verified ecDNAs using FISH in 30 cancer cell line, of which 9 cell lines are overlapped with the cell lines in our study. Therefore, we collected 10 ecDNAs from these nine cell lines which are verified by FISH in their study^{18,19}. Among these, eight were highly overlapped with the ecDNA regions detected by our ecDNA detection (Fig. R10f).

Figure R10. (a) Heatmap of ecDNA amplicon correlation in SF268 cell line. (b) The number of ecDNA region after the integration analysis using an unsupervised hierarchical clustering method. (c) Detection of ecDNA amplicons with junction site. (d) ecDNA regions validated by FISH experiments.

chromatin tracing

1. When they propagate cells, the scRNA seq clusterings change. That is not very surprising and could be caused by many factors including sampling variation. What controls are used?

Answer:

We apologize for the confusion. In this study, we utilized the genetic barcoding strategy (Cell-ID)

to stably mark the cell lineage (Fig. R11a, also as supplementary Fig. 7a in the manuscript). For this purpose, to assure each ‘ancestor’ cell was labeled with only one distinct barcode, we transfected 150,000 cells with a barcode library consisting of 230,000 distinct barcodes at a MOI of 0.1. Therefore, the chance of one cell getting more than one barcode and that of two cells getting a same barcode is negligible. From the ‘ancestor’ cell population, we seeded 200 cells for around 10x expanding. We then took cells at two time points (two weeks in between) for scRNA-seq. As a result, cells with the same Cell-ID were derived from the same ancestor. The cells were grouped into three clusters based on scRNA-seq at two time points in both cell lines (Fig. R11b, also as Fig. 7a and supplementary Fig. 7b in the manuscript). Surprisingly, based on scRNA-seq data, the cells derived from a same ancestor did not match to the same cluster (Fig. R11c, also as Fig. 7b and supplementary Fig. 7c in the manuscript). On one hand, at both time points, we observed that cells with the same barcode often were assigned to different clusters (Fig. R11c, also as Fig. 7c and supplementary Fig. 7c in the manuscript). On the other hand, even for those cells with the same barcodes assigned to only one cluster at time 1, the cells with these barcodes could be assigned to different clusters in time 2 (Fig. R11d, also as Fig. 7c and supplementary Fig. 7d). Take MDA-MB-231 as an example, cells with barcode 4 and barcode 23 were exclusively in cluster 0 at time point 1, but at time point 2, 21% of cells with barcode 4 and 35% of cells with barcode23 were in other clusters instead of cluster 0 (Fig. R11e). Taken together, we demonstrated that whereas the cellular transcriptome profile remains stable as a population, the transcriptome profile of individual cells is highly plastic and able to shift to different states, highlighting the dynamic nature of cellular heterogeneity. To make our points clearer, we rewrite this part in both Results and Methods section.

Figure R11. (a), The scheme of tracing experiment: cells were transfected with unique Cell-ID, then grown from 200 cells and analyzed in two different time points. (b), UMAP plot of MDA-MB-231 at T1 and T2. N represents cell numbers in different subgroups. (c), Venn diagram of barcodes in different subclusters of MDA-MB-231 at T1 and T2. (d), The distribution of unique barcodes in different subclusters at T1 and T2. Timepoint combined with the original cluster number was indicated, e.g. T1_0 represented cluster0 from T1. (e) The heatmap showing the similarity.

- 1 Kinker, G. S. *et al.* Pan-cancer single-cell RNA-seq identifies recurring programs of cellular heterogeneity. *Nat Genet* **52**, 1208-1218, doi:10.1038/s41588-020-00726-6 (2020).
- 2 Aibar, S. *et al.* SCENIC: single-cell regulatory network inference and clustering. *Nat Methods* **14**, 1083-1086, doi:10.1038/nmeth.4463 (2017).
- 3 Zhou, Y. *et al.* Metascape provides a biologist-oriented resource for the analysis of systems-level datasets. *Nat Commun* **10**, 1523, doi:10.1038/s41467-019-09234-6 (2019).
- 4 Darnell, J. E., Jr. Transcription factors as targets for cancer therapy. *Nat Rev Cancer* **2**, 740-749, doi:10.1038/nrc906 (2002).

- 5 Lombardi, O. *et al.* Pan-cancer analysis of tissue and single-cell HIF-pathway activation using a conserved gene signature. *Cell Rep* **41**, 111652, doi:10.1016/j.celrep.2022.111652 (2022).
- 6 Ma, S. *et al.* Chromatin Potential Identified by Shared Single-Cell Profiling of RNA and Chromatin. *Cell* **183**, 1103-1116 e1120, doi:10.1016/j.cell.2020.09.056 (2020).
- 7 Ghandi, M. *et al.* Next-generation characterization of the Cancer Cell Line Encyclopedia. *Nature* **569**, 503-508, doi:10.1038/s41586-019-1186-3 (2019).
- 8 Granja, J. M. *et al.* ArchR is a scalable software package for integrative single-cell chromatin accessibility analysis. *Nat Genet* **53**, 403-411, doi:10.1038/s41588-021-00790-6 (2021).
- 9 Schep, A. N., Wu, B., Buenrostro, J. D. & Greenleaf, W. J. chromVAR: inferring transcription-factor-associated accessibility from single-cell epigenomic data. *Nat Methods* **14**, 975-978, doi:10.1038/nmeth.4401 (2017).
- 10 Ma, L. *et al.* Tumor Cell Biodiversity Drives Microenvironmental Reprogramming in Liver Cancer. *Cancer Cell* **36**, 418-430 e416, doi:10.1016/j.ccell.2019.08.007 (2019).
- 11 Patel, A. P. *et al.* Single-cell RNA-seq highlights intratumoral heterogeneity in primary glioblastoma. *Science* **344**, 1396-1401, doi:10.1126/science.1254257 (2014).
- 12 Chen, Y. P. *et al.* Single-cell transcriptomics reveals regulators underlying immune cell diversity and immune subtypes associated with prognosis in nasopharyngeal carcinoma. *Cell Res* **30**, 1024-1042, doi:10.1038/s41422-020-0374-x (2020).
- 13 Xing, X. *et al.* Decoding the multicellular ecosystem of lung adenocarcinoma manifested as pulmonary subsolid nodules by single-cell RNA sequencing. *Sci Adv* **7**, doi:10.1126/sciadv.abd9738 (2021).
- 14 Wang, R. *et al.* Single-cell dissection of intratumoral heterogeneity and lineage diversity in metastatic gastric adenocarcinoma. *Nat Med* **27**, 141-151, doi:10.1038/s41591-020-1125-8 (2021).
- 15 Zhang, L. *et al.* Integrated single-cell RNA sequencing analysis reveals distinct cellular and transcriptional modules associated with survival in lung cancer. *Signal Transduct Target Ther* **7**, 9, doi:10.1038/s41392-021-00824-9 (2022).
- 16 Richards, L. M. *et al.* Gradient of Developmental and Injury Response transcriptional states defines functional vulnerabilities underpinning glioblastoma heterogeneity. *Nat Cancer* **2**, 157-173, doi:10.1038/s43018-020-00154-9 (2021).
- 17 Kumar, P. *et al.* ATAC-seq identifies thousands of extrachromosomal circular DNA in cancer and cell lines. *Sci Adv* **6**, eaba2489, doi:10.1126/sciadv.aba2489 (2020).
- 18 Deshpande, V. *et al.* Exploring the landscape of focal amplifications in cancer using AmpliconArchitect. *Nat Commun* **10**, 392, doi:10.1038/s41467-018-08200-y (2019).
- 19 Hung, K. L. *et al.* ecDNA hubs drive cooperative intermolecular oncogene expression. *Nature* **600**, 731-736, doi:10.1038/s41586-021-04116-8 (2021).

Reviewers' Comments:

Reviewer #1:

Remarks to the Author:

I thank the authors for comprehensively addressing my comments on the previous version - I feel that the manuscript is substantially improved and contains a wealth of data that will facilitate further research in this field.

Reviewer #2:

Remarks to the Author:

All my comments are addressed and I have no further comments.

Reviewer #3:

Remarks to the Author:

To address the concerns, the authors provide extensive clarifications and explanation of the methods and the data. The computational approaches are strong. However, a major new analysis of TFs has been done that raises concerns. This analysis raises some surprising findings regarding TP53. They identify enrichment for TP53 binding motifs on TP53 and show high level TP53 expression in some of the cells. However, TP53 is frequently mutated including with loss of function in cancer cell lines and tumor samples harboring ecDNA, making these findings very difficult to interpret. Perhaps I have misunderstood, but these data are very hard to understand.

Point-by-point response to the referees' comments

Reviewer #3 (Remarks to the Author):

To address the concerns, the authors provide extensive clarifications and explanation of the methods and the data. The computational approaches are strong. However, a major new analysis of TFs has been done that raises concerns. This analysis raises some surprising findings regarding TP53. They identify enrichment for TP53 binding motifs on TP53 and show high level TP53 expression in some of the cells. However, TP53 is frequently mutated including with loss of function in cancer cell lines and tumor samples harboring ecDNA, making these findings very difficult to interpret. Perhaps I have misunderstood, but these data are very hard to understand.

Answer: We appreciate the positive feedback from this referee and apologize for the confusion as the referee pointed out.

To identify the transcription factors (TFs) regulating the ecDNA oncogenes, we employed Metascape, which predicts the TF-target relationship based on the published data^{1,2}. These data encompass three types of TF-target regulatory modes: repression, activation, and complex. Moreover, the targets annotated here are not necessary under direct transcriptional regulation of the specific TF. Using Metascape, the upstream TF regulators are then predicted according to the degree of overlap between the annotated target genes of each TF and the input genes.

TP53 is a well-known tumor suppressor gene and is frequently mutated in advanced tumors^{3,4}. While losing the tumor suppressor activity, some p53 mutants gain new functions (GOF) to drive cancer progression, including promoting cell proliferation, migration, metastasis, and metabolism in various types of cancer⁵⁻⁸. Some oncogenes could be repressed by wild-type TP53 but activated by GOF TP53 mutants.

Using Metascape, we revealed that 15 out of 28 ecDNA oncogenes were regulated by TP53 (wildtype and/or GOF mutant) (Table R1^{9,10}) in five cell lines (Table R2). In MDA-MB231, p53-R280K is a gain-of-function mutant, which plays an important role in mediating the survival of MDA-MB231 cells^{3,11}. In this cell line, we found that the expression level of mutated P53 is positively correlated with that of downstream oncogenes, both of which are enriched in R0 (Fig R1a and b).

Figure R1. (a) The expression distribution of *TP53* in different scRNA-seq clusters of MDA-MB-231. P value was calculated by the Student's T-test. (b) The GSVAscore of gene sets targeted by the TF TP53 in different scRNA-seq clusters of MDA-MB-231. P value was calculated by the Student's T-test.

Table R1. TP53 status in different cell lines.

Cell line	P53 status	Mutation AA
SW480	p53mut	P309S, R273H
SNB75	p53mut	E258K
MDA-MB453	p53mut	Q331fs (Deletion)
MDA-MB231	p53mut	R280K
DLD-1	p53mut	S241F

Table R2. Target genes of TP53.

Target	Mode of Regulation by wildtype P53	References	Mode of Regulation by P53 GOF mutants	References
EZH2	Repression	12	Activation	13
CDK4	Complex		Activation	14

AKT1	Repression	15	Activation	16
STAT3	Repression	17	Activation	18
BRCA1	Repression	19	Activation	20,21
POLD1	Complex	22	Activation	23
PDGFRB	Activation	24	Activation	8
CCND1	Repression	25	Activation	26
XPO1	Repression	27	Activation	28
MET	Repression	29	Activation	30
MYC	Repression	31	Activation	32
DDB2	Activation	33	Activation	34
BRCA2	Repression	35	Activation	21
SMAD3	Complex	36	Activation	37
PML	Activation	38	Activation	39

Reference

- 1 Zhou, Y. *et al.* Metascape provides a biologist-oriented resource for the analysis of systems-level datasets. *Nat Commun* **10**, 1523, doi:10.1038/s41467-019-09234-6 (2019).
- 2 Han, H. *et al.* TRRUST v2: an expanded reference database of human and mouse transcriptional regulatory interactions. *Nucleic Acids Res* **46**, D380-D386, doi:10.1093/nar/gkx1013 (2018).
- 3 Kim, M. P. & Lozano, G. Mutant p53 partners in crime. *Cell Death Differ* **25**, 161-168, doi:10.1038/cdd.2017.185 (2018).
- 4 Li, T. *et al.* Tumor suppression in the absence of p53-mediated cell-cycle arrest, apoptosis, and senescence. *Cell* **149**, 1269-1283, doi:10.1016/j.cell.2012.04.026 (2012).
- 5 Dittmer, D. *et al.* Gain of function mutations in p53. *Nat Genet* **4**, 42-46, doi:10.1038/ng0593-42 (1993).
- 6 Olive, K. P. *et al.* Mutant p53 gain of function in two mouse models of Li-Fraumeni syndrome. *Cell* **119**, 847-860, doi:10.1016/j.cell.2004.11.004 (2004).
- 7 Freed-Pastor, W. A. *et al.* Mutant p53 disrupts mammary tissue architecture via the mevalonate pathway. *Cell* **148**, 244-258, doi:10.1016/j.cell.2011.12.017 (2012).
- 8 Weissmueller, S. *et al.* Mutant p53 drives pancreatic cancer metastasis through cell-autonomous PDGF receptor beta signaling. *Cell* **157**, 382-394, doi:10.1016/j.cell.2014.01.066 (2014).
- 9 Leroy, B. *et al.* Analysis of TP53 mutation status in human cancer cell lines: a reassessment. *Hum Mutat* **35**, 756-765, doi:10.1002/humu.22556 (2014).
- 10 Ghandi, M. *et al.* Next-generation characterization of the Cancer Cell Line Encyclopedia. *Nature* **569**, 503-508, doi:10.1038/s41586-019-1186-3 (2019).
- 11 Vogiatzi, F. *et al.* Mutant p53 promotes tumor progression and metastasis by the endoplasmic reticulum UDPase ENTPD5. *Proc Natl Acad Sci U S A* **113**, E8433-E8442, doi:10.1073/pnas.1612711114 (2016).
- 12 Tang, X. *et al.* Activated p53 suppresses the histone methyltransferase EZH2 gene. *Oncogene* **23**, 5759-5769, doi:10.1038/sj.onc.1207706 (2004).

- 13 Zhao, Y. *et al.* EZH2 cooperates with gain-of-function p53 mutants to promote cancer growth and metastasis. *EMBO J* **38**, doi:10.15252/embj.201899599 (2019).
- 14 Yang, D. *et al.* The over-expression of p53 H179Y residue mutation causes the increase of cyclin A1 and Cdk4 expression in HELF cells. *Mol Cell Biochem* **304**, 219-226, doi:10.1007/s11010-007-9503-9 (2007).
- 15 Oren, M. *et al.* Regulation of p53: intricate loops and delicate balances. *Ann N Y Acad Sci* **973**, 374-383, doi:10.1111/j.1749-6632.2002.tb04669.x (2002).
- 16 Yue, X. *et al.* Gain of function mutant p53 protein activates AKT through the Rac1 signaling to promote tumorigenesis. *Cell Cycle* **19**, 1338-1351, doi:10.1080/15384101.2020.1749790 (2020).
- 17 Lin, J. *et al.* Modulation of signal transducer and activator of transcription 3 activities by p53 tumor suppressor in breast cancer cells. *Cancer Res* **62**, 376-380 (2002).
- 18 Schulz-Heddergott, R. *et al.* Therapeutic Ablation of Gain-of-Function Mutant p53 in Colorectal Cancer Inhibits Stat3-Mediated Tumor Growth and Invasion. *Cancer Cell* **34**, 298-314 e297, doi:10.1016/j.ccell.2018.07.004 (2018).
- 19 Jiang, J. *et al.* p53-dependent BRCA1 nuclear export controls cellular susceptibility to DNA damage. *Cancer Res* **71**, 5546-5557, doi:10.1158/0008-5472.CAN-10-3423 (2011).
- 20 Peng, L., Xu, T., Long, T. & Zuo, H. Association Between BRCA Status and P53 Status in Breast Cancer: A Meta-Analysis. *Med Sci Monit* **22**, 1939-1945, doi:10.12659/msm.896260 (2016).
- 21 Ramus, S. J. *et al.* Increased frequency of TP53 mutations in BRCA1 and BRCA2 ovarian tumours. *Genes Chromosomes Cancer* **25**, 91-96, doi:10.1002/(sici)1098-2264(199906)25:2<91::aid-gcc3>3.0.co;2-5 (1999).
- 22 Li, B. & Lee, M. Y. Transcriptional regulation of the human DNA polymerase delta catalytic subunit gene POLD1 by p53 tumor suppressor and Sp1. *J Biol Chem* **276**, 29729-29739, doi:10.1074/jbc.M101167200 (2001).
- 23 Qin, Q. *et al.* Elevated expression of POLD1 is associated with poor prognosis in breast cancer. *Oncol Lett* **16**, 5591-5598, doi:10.3892/ol.2018.9392 (2018).
- 24 Yang, W., Wetterskog, D., Matsumoto, Y. & Funahashi, K. Kinetics of repression by modified p53 on the PDGF beta-receptor promoter. *Int J Cancer* **123**, 2020-2030, doi:10.1002/ijc.23735 (2008).
- 25 Rocha, S., Martin, A. M., Meek, D. W. & Perkins, N. D. p53 represses cyclin D1 transcription through down regulation of Bcl-3 and inducing increased association of the p52 NF-kappaB subunit with histone deacetylase 1. *Mol Cell Biol* **23**, 4713-4727, doi:10.1128/MCB.23.13.4713-4727.2003 (2003).
- 26 Donehower, L. A. *et al.* Integrated Analysis of TP53 Gene and Pathway Alterations in The Cancer Genome Atlas. *Cell Rep* **28**, 1370-1384 e1375, doi:10.1016/j.celrep.2019.07.001 (2019).
- 27 Golomb, L. *et al.* Importin 7 and exportin 1 link c-Myc and p53 to regulation of ribosomal biogenesis. *Mol Cell* **45**, 222-232, doi:10.1016/j.molcel.2011.11.022 (2012).
- 28 Deng, M. *et al.* XPO1 expression worsens the prognosis of unfavorable DLBCL that can be effectively targeted by selinexor in the absence of mutant p53. *J Hematol Oncol* **13**, 148, doi:10.1186/s13045-020-00982-3 (2020).
- 29 Hwang, C. I. *et al.* Wild-type p53 controls cell motility and invasion by dual regulation of

- MET expression. *Proc Natl Acad Sci U S A* **108**, 14240-14245, doi:10.1073/pnas.1017536108 (2011).
- 30 Muller, P. A. *et al.* Mutant p53 enhances MET trafficking and signalling to drive cell scattering and invasion. *Oncogene* **32**, 1252-1265, doi:10.1038/onc.2012.148 (2013).
- 31 Grinkevich, V. V. *et al.* Ablation of key oncogenic pathways by RITA-reactivated p53 is required for efficient apoptosis. *Cancer Cell* **15**, 441-453, doi:10.1016/j.ccr.2009.03.021 (2009).
- 32 Huang, X. *et al.* A novel PTEN/mutant p53/c-Myc/Bcl-XL axis mediates context-dependent oncogenic effects of PTEN with implications for cancer prognosis and therapy. *Neoplasia* **15**, 952-965, doi:10.1593/neo.13376 (2013).
- 33 Navaraj, A., Mori, T. & El-Deiry, W. S. Cooperation between BRCA1 and p53 in repair of cyclobutane pyrimidine dimers. *Cancer Biol Ther* **4**, 1409-1414, doi:10.4161/cbt.4.12.2378 (2005).
- 34 He, Y. H. *et al.* ERalpha determines the chemo-resistant function of mutant p53 involving the switch between lincRNA-p21 and DDB2 expressions. *Mol Ther Nucleic Acids* **25**, 536-553, doi:10.1016/j.omtn.2021.07.022 (2021).
- 35 Andres, J. L. *et al.* Regulation of BRCA1 and BRCA2 expression in human breast cancer cells by DNA-damaging agents. *Oncogene* **16**, 2229-2241, doi:10.1038/sj.onc.1201752 (1998).
- 36 Guan, B., Wang, T. L. & Shih le, M. ARID1A, a factor that promotes formation of SWI/SNF-mediated chromatin remodeling, is a tumor suppressor in gynecologic cancers. *Cancer Res* **71**, 6718-6727, doi:10.1158/0008-5472.CAN-11-1562 (2011).
- 37 Adorno, M. *et al.* A Mutant-p53/Smad complex opposes p63 to empower TGFbeta-induced metastasis. *Cell* **137**, 87-98, doi:10.1016/j.cell.2009.01.039 (2009).
- 38 de Stanchina, E. *et al.* PML is a direct p53 target that modulates p53 effector functions. *Mol Cell* **13**, 523-535, doi:10.1016/s1097-2765(04)00062-0 (2004).
- 39 Haupt, S. *et al.* Promyelocytic leukemia protein is required for gain of function by mutant p53. *Cancer Res* **69**, 4818-4826, doi:10.1158/0008-5472.CAN-08-4010 (2009).

Reviewers' Comments:

Reviewer #3:

Remarks to the Author:

The authors have done a good job of addressing the final concern.